

# Source apportionment of PM₂.₅ in Shanghai based on hourly molecular organic markers and other source tracers

Rui Li[a+], Qiongqiong Wang[b+], Xiao He[c], Shuhui Zhu[c,d], Kun Zhang[a], Yusen Duan[e], Qingyan Fu[e], Liping Qiao[d], Yangjun Wang[a], Ling Huang[a], Li Li[a*], and Jian Zhen Yu[b,c*]

[a] School of Environmental and Chemical Engineering, Shanghai University, Shanghai, 200444, China

[b] Department of Chemistry, [c] Division of Environment and Sustainability, Hong Kong University of Science and Technology, Hong Kong, China

[d] State Environmental Protection Key Laboratory of the Cause and Prevention of Urban Air Pollution Complex, Shanghai Academy of Environmental Sciences, Shanghai, 200233, China

[e] Shanghai Environmental Monitoring Centre, Shanghai, 200235, China

[+]*These two authors contributed equally to this work.*

[*]*Correspondence to*: Li Li (Lily@shu.edu.cn) and JianZhen Yu (jian.yu@ust.hk)

**Abstract**

Identification of various sources and quantification of their contributions are a necessary step to formulating scientifically sound pollution control strategies. Receptor model is widely used in source apportionment of fine particles. However, most of the previous studies are based on traditional filter collection and lab analysis of aerosol chemical species (usually ions, elemental carbon (EC), organic carbon (OC) and elements) as inputs. In this study, we conducted robust online measurements of a range of organic molecular makers and trace elements, in addition to the major aerosol components (ions, OC and EC), in urban Shanghai in the Yangtze River Delta region, China. The large suite of molecular and elemental tracers, together with water-soluble ions, OC and EC, provide data for establishing measurement-based source apportionment methodology for PM₂.₅. We conducted source apportionment using positive matrix factorization (PMF) and compared PMF solutions with molecular makers added (i.e. MM-PMF) and those without organic markers. MM-PMF identified 11 types of pollution sources, with biomass burning, cooking and secondary organic aerosol (SOA) as the additional sources identified. The three sources accounted for 4.9%, 2.6% and 14.7% of the total PM₂.₅ mass, respectively. During the whole campaign, the secondary source is an important source of atmospheric pollution, the average contribution of secondary pollution sources is as high as 63.8% of the total PM₂.₅ mass. Grouping different sources to secondary and primary, we note that SOC and POC contributed 45.1% and 54.9%, respectively. It is worth noting that the contribution of cooking to PM₂.₅ mass only account for 2.6%, but it contributed to 10.7% of OC. Episodic analysis indicated that secondary nitrate was always the main cause of PM₂.₅ pollution, while during non-episodic hours, vehicle exhaust made a significant





contribution. Through the application of the above-mentioned techniques to the Yangtze River Delta, more insights are
gained on the sources, formation mechanism and pollution characteristics of PM$_{2.5}$ in this region.
**1. Introduction**
In recent years, with the increasingly prominent problem of air quality, more and more attention has been paid to the
research of air pollution, which focuses on the study of atmospheric particulate matter (PM), especially fine particulate
matter (PM$_{2.5}$) (Chen et al., 2007; Zhang et al., 2009a). The study of chemical composition of atmospheric PM$_{2.5}$ is to help
understand the source, formation mechanism, and environmental effects. PM$_{2.5}$ pollution reduces atmospheric visibility
(Chow et al., 2004) and exposure to PM$_{2.5}$ is positively correlated with adverse health effects (Nel, 2005; Lippmann et al.,
2009; Mimura et al., 2014; Liu et al., 2016; Jimenez et al., 2009). PM$_{2.5}$ in the atmosphere also affects the radiation balance
on the ground by scattering or absorbing solar radiation and changing the properties of clouds, which has an impact on
global climate (Foley et al., 2010; Ramanathan et al., 2001; Kanakidou et al., 2005). Therefore, PM$_{2.5}$ pollution has become
the primary control target to improve air quality in urban environments. Studying the composition of PM$_{2.5}$ and identifying
its sources have practical significance for the understanding of PM$_{2.5}$ pollution characteristics and environmental effects.
The analysis of the sources is of great significance for PM$_{2.5}$ emission reduction and pollution control (Huang et al.,
2014; Wang et al., 2017). Generally, the sources of pollution can be qualitatively assessed according to presence of chemical
components characteristic to specific sources. In addition, receptor models can be used to further identify and quantify the
sources of atmospheric PM$_{2.5}$ (Hopke, 2016; Jaeckels et al., 2007; Lee et al., 2008; Sofowote et al., 2014;). Compared with
other methods, Positive Matrix Factorization (PMF) (Paatero & Tapper, 1994) does not need to input the source profiles,
and we can simultaneously analyze the source profiles and contributions of various sources. Hence, PMF has been widely
used in the source analysis of PM$_{2.5}$ all over the world, for example, Beijing (Song et al., 2006; Yao et al., 2016), Shanghai
(Wang et al., 2018; Wang et al., 2013; Feng et al., 2013; Feng et al., 2012; Wang et al., 2015; Wang et al., 2014; Zhao et
al., 2015; Qiao et al., 2016), Hong Kong (Hu et al., 2010; Yuan et al., 2006), New York (Zhou et al., 2019; Masiol et al.,
2017), and other regions (Ulbrich et al., 2009; Fang et al., 2015).
High time resolution measurements are inherently advantageous to the source analysis, because they are able to
capture the diurnal variations of the main source activities (such as vehicle exhaust) and secondary formation processes.
High time resolution data provide opportunities to study short-term source distributions. There are limited number of source
apportionment studies exploring the combined high-time resolution data set including various aerosol components such as
trace elements (Wang et al., 2018). Online-measurement based source apportionment studies in the past, so far, are mainly



based on PM$_1$ Aerodyne aerosol mass spectrometer (AMS) or Aerosol chemical speciation monitoring (ACSM) (Al-
Naiema et al., 2018). AMS or ACSM provide data on ion fragments, which only retain partial information for their parent
molecules. Multiple parent molecules could lead to the same fragments during the ionization process in AMS or ACSM,
which introduce ambiguity in relying on fragment ions for source differentiation. Accurate quantitative monitoring of
atmospheric organic matter on the molecular level is critical to source analysis.
Shanghai is the financial center of China, with a large population and a total area of 6,340 km$^2$. Past source
apportionment studies of PM$_{2.5}$ in Shanghai were either based on offline filters using receptor models (Du et al., 2017;
Chang et al., 2018), or emissions with numerical models (Li et al., 2015; Shu et al., 2019; Li et al., 2019; Feng et al., 2019).
The past receptor modeling studies relied on chemical composition data derived from 24-h time-integrated filter samples
followed by off-line laboratory analysis. The off-line nature severely limits its utility in addressing episodic pollution events
and in providing data to assess emission-based model evaluation of pollution sources and regional transport. Some
researchers have conducted online PM$_{2.5}$ source apportionment, however, previous studies were mainly using the traditional
aerosol species as inputs (Wang et al., 2018). Organic matter constitutes a considerable share of PM$_{2.5}$, while the online
analytical techniques used in the past were not suitable for describing this fraction in full.
In this study, online monitoring of atmospheric PM$_{2.5}$ compositions, including inorganic ions, organic carbon (OC),
elemental carbon (EC), trace elements and organic molecular markers, was conducted in Shanghai from November 9 to
December 3, 2018. The purpose is to use the detailed high-time resolution speciation data (especially organic molecular
markers) to identify the sources of PM$_{2.5}$ based on molecular-marker based PMF. This study gives insights into more
detailed source contributions, changes of sources and effects of the air pollution control strategies.
**2. Methodology**
**2.1 Online measurement**
We conducted an observation for PM$_{2.5}$ and its major chemical compositions (including inorganic ions, OC, EC,
elements, and organic tracers) from November 9 to December 3, 2018. The organic tracers were measured in Shanghai
Academy of Environmental Sciences, which is a reprehensive site for the urban city. The inorganic ions, OC/EC, and
elements were measured in Shanghai Pudong Environmental Monitoring Station, which is also a typical site for the urban
city. Monitoring site locations are shown in Figure 1. The two sites are 12 km apart. We combined these data in order to
obtain a more comprehensive chemical characterization of the urban PM$_{2.5}$ air pollution situation.

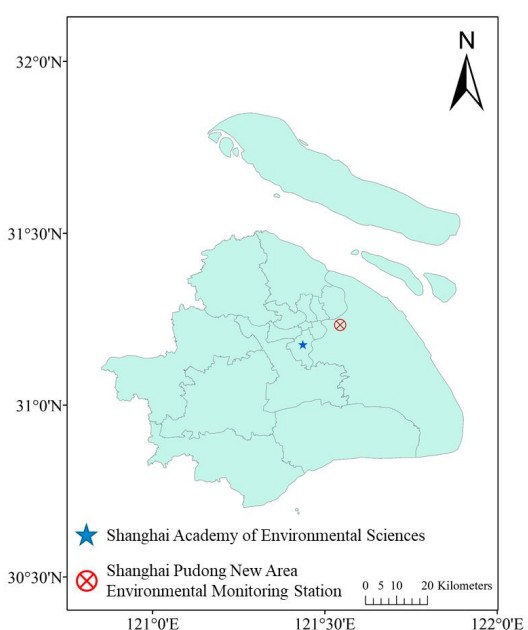


**Figure 1. Locations of the two sampling sites in Shanghai, China**

The concentration of $PM_{2.5}$ was measured by an online beta attenuation particulate monitor (FH 62 C14 series, Thermo

Fisher Scientific) (Qiao et al., 2014). Carbonaceous materials (OC and EC) were monitored by a semi continuous OC/EC
analyzer (model RT-4, Sunset Laboratory, Tigard, OR, USA) (Nicolosi et al., 2018; Zhang et al., 2017). The water-soluble
inorganic ions were measured by Monitor for Aerosols and Gases (MARGA, Model ADI 2080, Applikon Analytical B.V.)
(Makkonen et al., 2012; Griffith et al., 2015). Concentrations of 22 elements in $PM_{2.5}$ were measured by an ambient
elemental monitor (Xact 625 Ambient Continuous Multi-metals Monitor, Cooper Environmental Services, Tigard, OR,
USA) using nondispersive X-ray fluorescence (XRF) analysis (Battelle, 2012; Jeong et al., 2019). Quantification of hourly
speciated organic compounds were achieved using a Thermal desorption Aerosol Gas Chromatograph (TAG) (Williams et
al., 2014; Zhao et al., 2013a; Isaacman et al., 2014). The operation details and data quality are described in a separate paper
(Wang et al., 2019). Figure S1 shows the comparison of reconstructed and measured $PM_{2.5}$ mass for samples collected for
this study (Wang et al., 2016; Huang et al., 2014). The meteorological parameters are from the open data at Hongqiao
airport (available at http://www.wunderground.com).
**2.2 PMF receptor model**

PMF is a bilinear factor analysis method, which is widely used to identify pollution sources and quantify the



contribution of source sectors to the concentration of ambient air pollutants at receptor sites, with an assumption of mass
conservation and a chemical mass balance between emission source and receptors. In this study, the Environmental
Protection Agency (EPA) PMF version 5.0 (Norris et al., 2014) was applied to perform the analysis. PMF decomposes the
measured data matrix, $X_{ij}$, into a factor profile matrix, $f_{kj}$, and a factor contribution matrix, $g_{ik}$, (Eq 1):
$x_{ij} = \sum_{k=1}^{p} g_{ik} f_{kj} + e_{ij}$ (1)
$Q = \sum_{i=1}^{n} \sum_{j=1}^{m} (e_{ij}/u_{ij})^2$ (2)

In eq 1, $X_{ij}$ is the measured ambient concentration of target pollutants; $g_{ik}$ is the source contribution of the $k^{th}$ factor

to the $i_{th}$ sample, and $f_{kj}$ is the factor profile of the $j_{th}$ specie in the $k_{th}$ factor; $e_{ij}$ is the residual concentration for each data
point. PMF seeks a solution that minimizes an object function Q (Eq 2), based on the uncertainties of each observation $u_{ij}$.
The user provides the $u_{ij}$ for each data point. The selection of the best factor in this study and the error estimation diagnostics
for each model result are described in the supplement information (Figure S2, Figure S3, Table S1, Table S2).

PMF model assumes that the quantity of the input species is conserved, and the source profile is unchanged. In order

to minimize the impact of organic matter degradation on the deviation of mass conservation hypothesis, organic species
with low volatility and low reactivity are selected as input. The requirement of constant source profiles is not strictly met
when the receptor model is applied to measurement data covering a long duration (e.g., months or longer). However,
understanding/progress can be achieved despite the non-strict adherence to the requirements of the constant source profile.
The source profile parsed by PMF can be viewed as the averaged profile over the entire sampling period. In an atmospheric
environment, both primary organic aerosol (POA) and secondary organic aerosol (SOA) have the problem of changing
source profiles. Therefore, it is necessary and vital to obtain high time resolution data, preferably several hours for a sample
of data or shorter, as an input file for PMF model. The input files in this study are hourly data and the time span of whole
campaign is less than one month. As such, the source type information will not change significantly.

In this study, a total of 289 samples has been collected. The hourly chemical species selected as input to PMF model

contain 13 elements, 4 inorganic species, the carbon component (OC and EC), organic markers (including polycyclic
aromatic hydrocarbons (PAHs) and anhydro sugars, etc.), and $PM_{2.5}$ mass concentration. Traditional PMF (PMFt, "t" refers
to traditional) which include only $PM_{2.5}$ mass, elements, inorganic ions, OC and EC as inputs, and molecular marker based-
PMF (MM-PMF) (Al-Naiema et al., 2018; Wang et al., 2017; Zhang et al., 2009b) with organic markers added as inputs
on the basis of above species were performed separately to do source apportionment.

The uncertainty of each data point is calculated according to Eq 3:





$u_{ij} = \sqrt{(x_{ij} \times EF)^2 + (\frac{1}{2} \times MDL)^2}$     (3)
where MDL is the method detection limit and EF is the error fraction determined by the user and associated with the
measurement uncertainty. The concentration data of species below the detection limit were replaced by 1/2 of the MDL,
and the $u_{ij}$ was calculated by 5/6 of the MDL. For the concentration data of missing species, the missing value is the
geometric average value of the concentration of this species, and its $u_{ij}$ is four times the geometric average value.
**2.3 Trajectory analysis**
The backward trajectory analysis is a useful tool to identify the influence of air mass path on PMF resolved
sources. 36-h duration backward trajectories arriving at an altitude of 100 m above ground level (AGL) over the site were
calculated using the NOAA HYSPLIT model (https://ready.arl. noaa.gov/HYSPLIT.php), deploying the 0.5° Global Data
Assimilation System (GDAS) meteorological data. The sampling days were then classified into four clusters according to
the geographical origin and movement process of the trajectories, i.e., two oceanic trajectories, and two continental
trajectories.
**3. Results and discussion**
The pollution episodes occurred mostly in winter, due to adverse atmospheric conditions (such as more frequent
stagnation of atmospheric movement) and enhanced the impact on air quality from local and regional emissions. The
hourly meteorological parameters and $PM_{2.5}$ concentration during the monitoring time is shown in Fig. 2. The concentration
levels of the major species measured are provided in Table 1. The average temperature was 14°C, the relative humidity
(RH) was 79.9%, the wind speed (WS) was 3m/s during the campaign. The average $PM_{2.5}$ concentrations were 46.3±33.8
$\mu g/m^3$, with organic matter contributing to 19.6% of the total mass. Nitrate, sulfate, and ammonium contributed to 32.0%,
16.5%, and 16.2% of $PM_{2.5}$, respectively. Measured total elements account for 3.5% of $PM_{2.5}$ mass on average.



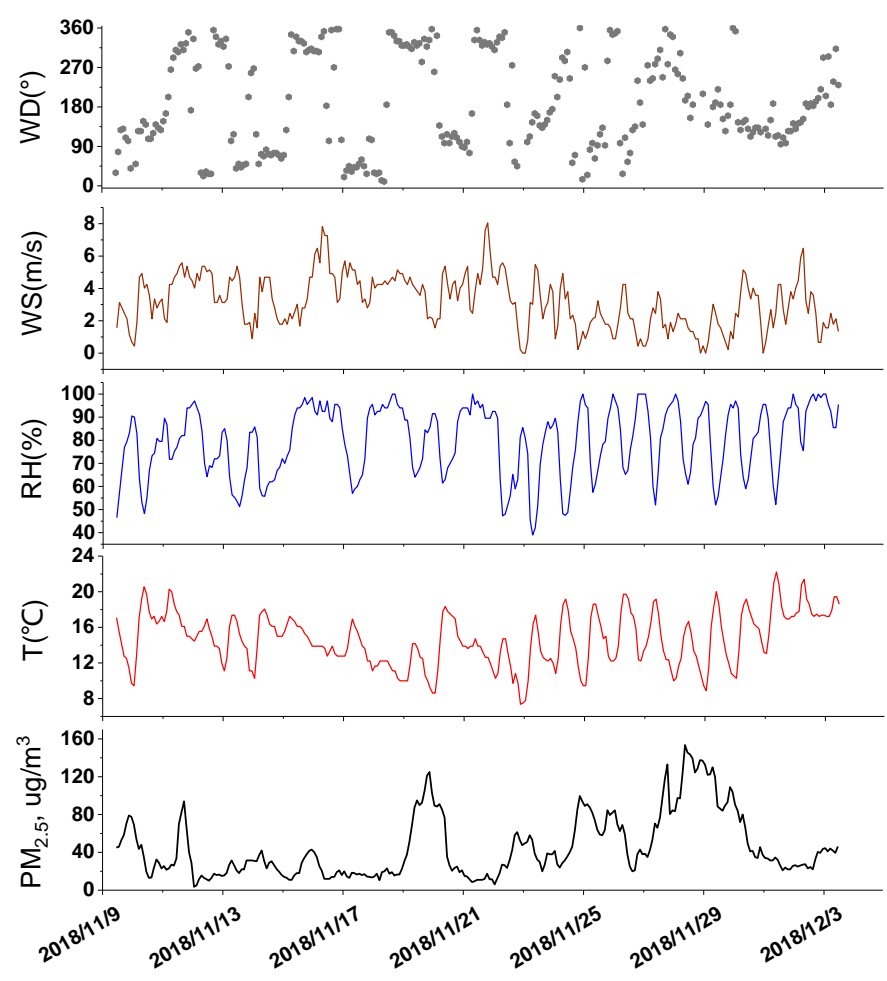

**Figure 2. Time series of meteorological parameters and PM$_{2.5}$ during the field campaign.**



**Table 1. Measured PM$_{2.5}$ major components (μg/m³) used in the PMF analysis in this study**

| Compound | Average | Stdev |
|---|---|---|
| PM$_{2.5}$ | 46.3 | 33.8 |
| Cl$^-$ | 0.78 | 0.52 |
| Nitrate | 14.81 | 15.12 |
| Sulfate | 7.65 | 4.31 |
| Ammonium | 7.47 | 6.25 |
| EC | 1.59 | 1.13 |
| OC | 6.48 | 2.79 |
| As | 0.006 | 0.005 |
| Ba | 0.024 | 0.017 |
| Ca | 0.137 | 0.104 |
| Cr | 0.004 | 0.005 |
| Cu | 0.012 | 0.008 |
| Fe | 0.445 | 0.627 |
| K | 0.381 | 0.196 |
| Mn | 0.065 | 0.069 |
| Ni | 0.004 | 0.003 |
| Pb | 0.025 | 0.026 |
| Si | 0.421 | 0.322 |
| V | 0.0031 | 0.0029 |
| Zn | 0.114 | 0.099 |


**Table 2. Abundance and naming of measured organic tracers (ng/m³) used in the MM-PMF Analysis**

| Naming | Grouping | Average | Stdev |
|---|---|---|---|
| PAHs252 | benzo[b]fluoranthene, benzo[k]fluoranthene, benzo[e]pyrene, and benzo[a]pyrene | 1.44 | 1.43 |
| PAHs276 | benzo[ghi]perylene, and indeno[1,2,3-cd]pyrene | 0.559 | 0.529 |
| C$_{6-8}$ DICAs | Adipic acid, Pimelic acid, and Suberic acid | 17.45 | 18.46 |
| C$_9$-acids | 9-Oxononanoic acid, and Azelaic acid | 9.25 | 6.46 |
| SFAs | Palmitic acid, and Stearic acid | 71.57 | 60.86 |
| Mannosan | | 1.54 | 1.51 |
| Levoglucosan | | 45.91 | 39.17 |
| OHBAs | 3-hydroxybenzoic acid, and 4-hydroxybenzoic acid | 1.05 | 0.85 |
| α-pinT | Pinic acid, and 3-methyl-1,2,3-butanetricarboxylic acid | 21.05 | 19.22 |
| DHOPA | 2,3-dihydroxy-4-oxopentanoic acid | 3.93 | 4.92 |
| Phthalic acid | | 9.13 | 10.28 |

**3.1 PM$_{2.5}$ source apportionment**
In this study, PMF source analysis was conducted in two scenarios. They were MM-PMF with organic tracers and





PMFt without organic tracers included and the results were compared in detail. The abundance and nomenclature of the
organic tracers used are shown in Table 2. The priority input species for PMF analysis are those with high abundance and
characteristics of different sources, especially organic compounds with lower volatility and lower reactivity were selected
as input for MM-PMF. Highly correlated organic species (R> 0.8), indicating common sources, are clustered together to
reduce the number of species and to avoid collinearity problems in MM-PMF (Wang et al., 2017).
**3.1.1 MM-PMF results**
In PMF, the optimal number of factors is a compromise between identifying factors with the best physical explanations
and achieving a sufficiently good fit for all species. In PMF solutions of too few factors, different sources are combined
together, the resolved sources cannot fully explain the individual species; while too many factors may split one source into
multiple uninterpretable factors. Initially, 7 to 14 factors were tested, and the final factor numbers were determined by
examining the change in $Q/Q_{exp}$. Finally, the 11-factor solution for MM-PMF was selected as it gave the most reasonable
factor profiles (Figure S2). Table S1 shows the summary of error estimation diagnostics from BS, DISP and BS-DISP for
MM-PMF.The summary of the model performance for individual input species for the 11-factor solution in MM-PMF are
given in Table S3. Nevertheless, the base run results still have certain degrees of factor mixing. As the source specific
tracer compounds have similar temporal variations and the diversity of species components contained in the source,
chloride, sulfate and certain metal elements were found in different PMF-resolved profiles.
Factor recognition was based on the highest loaded species of each factor. The factor profiles of the 11-factor solution
are shown in Fig. 3. Secondary nitrate factor (F1) was identified by high concentrations of nitrate and ammonium, the
distribution of ammonium and nitrate in F1 is 48.1% and 66.2% respectively. The secondary sulfate factor (F2) is
characterized by the highest loading of sulfate (35.3%). In addition, ammonium in the secondary sulfate factor distribution
ranks the second among the eleven factors, less than F1. A small amount of organics acids and PAHs also appear in the two
factors. OC contribution from the secondary sulfate, secondary nitrate, and SOA factors were assumed as secondary OC
(SOC), whereas OC from the other factors was assumed to be primary OC (POC). The secondary sulfate factor contributes
the most to SOC, which can be seen from the correlation of the respective species with OC. Percentage contribution of
various source factors to $PM_{2.5}$ and to OC based on MM-PMF show in Fig.4. F1 and F2 contributed 33.5% and 15.6% to
total $PM_{2.5}$ mass, respectively. The SOC associated with secondary nitrate and sulfate factors accounted for 2.68 μg/m³
(41.3%) on average across the whole observation period. The diurnal variation of F1 showed much higher contributions at
nighttime; while for F2, no obvious diurnal contrast was observed. F1 has the medium and highest correlation with $NO_x$



(R =0.54) and CO (R = 0.79) while F2 showed moderate correlations with $SO_2$ (R = 0.32) and CO (R =0.36) (Table S4).
These results suggest that F1 may represent condensation of oxidation products of local emissions in the nighttime plus
regional transportation. Sun et al. (2006) observed increased sulfate formation under high RH in winter in urban city.
During the observation period, the RH is high during the day and night, and the daytime temperature is higher than the
nighttime. Therefore, the combined effects of aqueous phase oxidation and daytime photochemical reactions, leading to
the not obvious daily variation of the secondary sulfate factor.

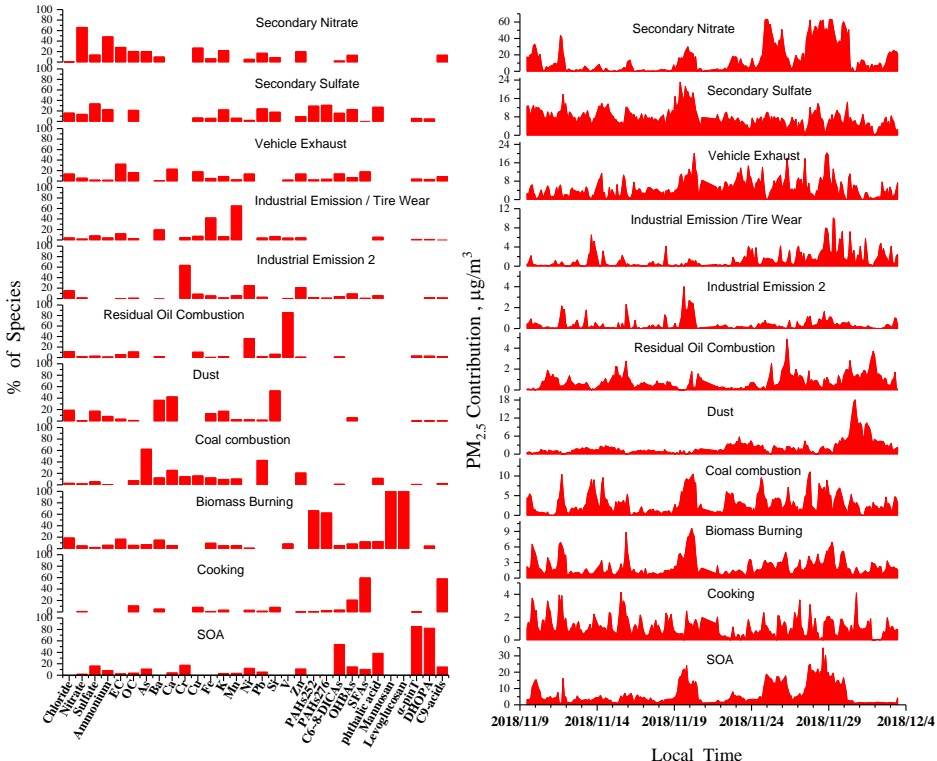


**Figure 3. Resolved factor profiles (left) and factor contributions (right) in eleven-factor solution in MM-PMF**
The third factor (F3) has a high abundance of EC, OC, Ca, Cu and also contains some organic tracers (PAHs and
organic acids, etc.), contributing to 11.3% of the total $PM_{2.5}$ mass on average. The source of vehicle exhaust contributes the
most to POC, accounting for 16.2% to OC. Vehicle exhaust is an important source for carbonaceous species, the presence
of Cu may originate from both fuel/lubricant combustion and brake abrasions (Adachi and Tainosho, 2004; Pant and
Harrison, 2013), and the element Ca may be derived from road dust. The influence of vehicle exhaust on this factor is
supported by the peak hours at 7:00-9:00 am and 5:00-7:00 pm in daily variation (Fig. 5). In addition, the high correlation



with NOx (R=67) and CO (R=58) indicate that vehicle exhuast have a significant impact on this factor. The higher nighttime
than daytime contribution of this factor may suggest influence from the planetary boundary layer height variation. In the
daytime, higher boundary height leads to more vertical mixing of the pollutants and facilitates dispersion, while the stagnant
nighttime atmosphere easily accumulates pollutants (Liu and Liang, 2010).

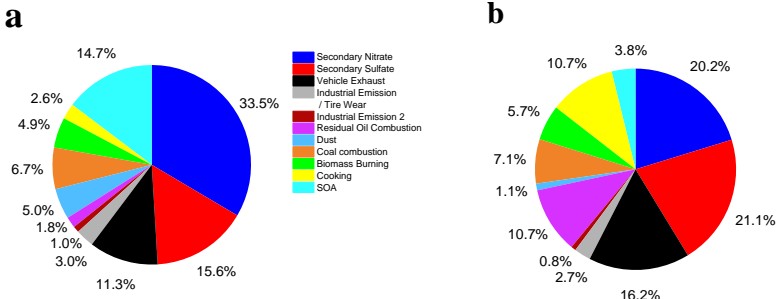

**Figure 4. Percentage contribution of various source factors to PM$_{2.5}$ (a) and to OC (b) based on MM-PMF.**
Factor 4 contains high loads of Fe and Mn as source tracers, in addition, it contains Ba as well. Most of these metal
elements come from industries related to steel production. Although metallic elements are not exclusive tracers of industrial
emissions, there are no other characteristic elements or compounds to track industrial emissions.In addition, the diurnal
variation of F4 is similar to that of F3, which may be due to the tire wear in the morning and evening peak. Manganese-
ferro, Zn, Cu and other elements have also been reported to be related to tire wear (Pant and Harrison, 2013), and Wang et
al. (2018) also revealed this in their research on online data. While F2 showed moderate correlations with SO$_2$ (R = 0.52) ,
CO(R=0.35)and NO$_X$(R=49),therefore, F4 is considered as the industrial and Vehicle emission source.The factor
contribution to total PM$_{2.5}$ mass and OC was minor, only 3.0% and 2.7%, respectively. Analysis of membrane samples in
the area did not reveal this phenomenon (Du et al., 2017; Huang et al., 2014; Qiao et al., 2016), suggesting the benefit of
the online high-resolution measurement.The fifth factor (F5) is characterized by high concentrations of Cr and Ni, which
are often used in industrial processes such as plating, tanning, and metallurgy (Karar et al., 2006; Borai et al., 2002). This
factor showed best correlation with CO (R = 59). No diurnal variation was observed. The factor contribution to total PM$_{2.5}$
and OC mass were minor, only 1.0% and 0.8%. The residual oil combustion factor (F6) was identified with V and Ni as
tracers, of which V is often used as a tracer for residual oil combustion. The contribution of this factor mainly comes from
ship transportation (Zhao et al., 2013b). The diurnal contribution at night was greater than that during daytime. The factor,
residual oil combustion, were minor contributors to PM$_{2.5}$, accounting for 1.8%, but the contribution (10.7%) of this factor
to OC is not negligible. The dust factor (F7) was distinguished by crustal elements Ca, Si, and Ba. The diurnal variation of



this factor showed a broad peak during the daytime, and negative correlation with RH (R=-0.26), suggesting an influence
from meteorological conditions. The factor (dust) contribution to total $PM_{2.5}$ and OC mass were 5.0% and 1.1%,
respectively.

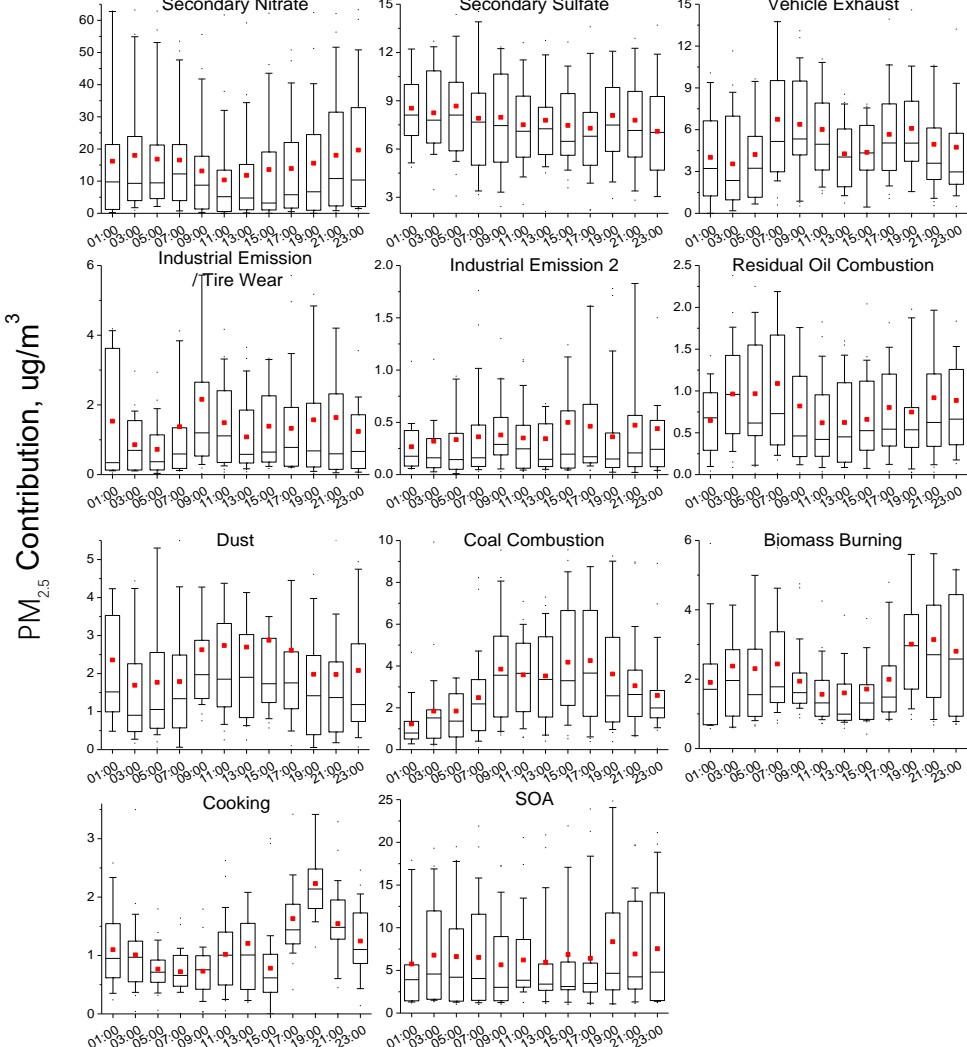


**Figure 5. Diurnal variation of various source factors based on MM-PMF.** (25th and 75th percentile boxes, 10th and 90th

percentile whiskers; lines inside the boxes represent the hourly median and the red points represent the hourly mean)

The coal combustion factor (F8) contains high loading of metals As and Pb, accounting for 6.7 % of $PM_{2.5}$ mass on
average and 7.1% to OC, it is mostly associated with coal combustion (Chen et al., 2013). Good correlations with $SO_2$ (R





= 59) and CO (R =56) further support the identification of this factor. Based on analysis of MM-PMF, there is no specific
organic tracers such as PAHs present in the source. The results are different from those of Wang et al. (2017) and Yu et al.
(2016), which are related to regional differences in source classes.

The F9 and F10 were resolved, namely, biomass burning identified by levoglucosan and mannosan; cooking aerosol

by SFAs (Palmitic acid, and Stearic acid) and $C_9$-acids (9-oxononanoic acid, and azelaic acid). Levoglucosan is uniquely
emitted by biomass burning activities (Engling et al., 2006; Feng et al., 2013). Using Levoglucosan to indicate biomass
burning may avoid the ambiguity of using K to determine the biomass burning source, which improves the accuracy of
source analysis. This factor, contributed 4.9% and 5.7% of total $PM_{2.5}$ and OC mass on average, respectively. In addition,
biomass burning contains high-loading of five-ring and six-ring PAHs that are considered to be derived from a mixed
combustion source (including coal combustion, biomass burning, etc.). It is worth noting that the contribution of cooking
to $PM_{2.5}$ mass only account for 2.6%, but it contributed 10.7% to OC. In addition, by analyzing the diurnal variation of
biomass burning and cooking, the biomass burning emission at night is greater than at daytime, while cooking has an
obvious peak value during 5:00-9:00 pm, which is consistent with the local dining consumption habits in Shanghai.

SOA was identified by toluene SOA tracer (2,3-dihydroxy-4-oxopentanoic acid) , α-pinene SOA tracers (pinic acid

and 3-methyl-1,2,3-butanetricarboxylic acid) and phthalic acid. It is a major source of $PM_{2.5}$, accounting for 14.7%. SOC
from the SOA factor accounted for 0.25 μg/m³ (3.8%) on average to OC. SOA related to toluene was considered as
anthropogenic SOA, and SOA containing pinic acid was considered as biogenic SOA. It was found that there was a similar
daily changes between SOA and secondary nitrate factor, indicating some commonality in their formation processes. Many
studies have documented the enhancement of biogenic SOA production by anthropogenic species through creating a more
acidic environment in the aerosol (Jang et al., 2002; Wang et al., 2017).

Overall, the source apportionment results showed that secondary sources accounted for 63.9% of the total $PM_{2.5}$ mass

from MM-PMF, and the contribution of secondary nitrate to $PM_{2.5}$ was greater than that of secondary sulfate. SOA is the
third largest source of $PM_{2.5}$, followed by vehicle exhaust, which is the largest source of primary sources. The SOC
associated with secondary nitrate, secondary sulfate and SOA factors accounted for 45.1% on average of the total OC mass
across all the study period. The high loading of SOC in the secondary nitrate and sulfate factor compared with the SOA
factor may indicate the potential mix of the SOA in secondary sulfate factor due to the limited organic tracers included.
POC accounts for 3.55 μg/m³ (54.8%) of the total OC. And vehicle exhaust contributes the most to POC. MM-PMF gives
more detailed allocation of $PM_{2.5}$ to more accurate source factors.



### 3.1.2. Back Trajectory Analysis of MM-PMF-Resolved Sources


Previous studies based on source models have shown the importance of regional transportation and local emissions to
Shanghai haze events. Wang et al. (2014) identified two types of haze events in November 2010: local emission plays a
dominant role in the case of weak wind (WS < 0.5m /s), while in the case of stroke (~ 2m /s), regional transport from the
upwind region contributes the most. Li et al. (2015) found that local emission (~50%) in Shanghai in January 2013 was the
most significant factor causing pollution. Wang et al. (2018) studied with online high-resolution data in December 2014,
and found that local emission had an obvious impact on most pollution sources, while dust emission and coal combustion
had a greater impact on the region.

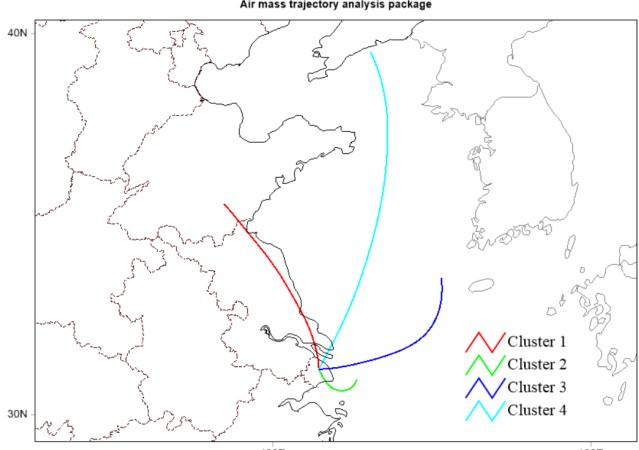


**Figure 6. Clustering of air mass trajectories during observation**
During the observation period, all the air mass trajectories at the receptor are shown in Figure S4. Fig.6 and table S5
show the clustering results of all air mass trajectories and the average $PM_{2.5}$ concentration measured at the sampling time
represented by each cluster. These four clusters accounted for 16.1%, 40.7%, 14.3% and 28.9% of the total trajectory,
respectively. It can be seen from Fig.7 that the composition varies with the source of the air mass. The average concentration
of modeled $PM_{2.5}$ of each cluster analyzed by the model was 53.7 $\mu g/m^3$, 61.5 $\mu g/m^3$, 32.6 $\mu g/m^3$ and 23.0 $\mu g/m^3$,
respectively. By comparing the four clusters, nitrate was the most important reason for PM pollution in Shanghai.
The effect of pollutant accumulation caused by air mass transport represented by cluster 1 and 2 on the average $PM_{2.5}$
concentration is significantly greater than the effect of cluster 3 and 4 air mass transport on $PM_{2.5}$. The average
concentration of $PM_{2.5}$ in Cluster 1 is slightly larger than that of Cluster 2. According to the length and source of the
trajectories of the respective clusters, cluster 1 is greatly affected by the urban transmission in North China. Coal





combustion, biomass burning, vehicle exhaust and secondary sulfate contribution to PM$_{2.5}$ concentration are greater than
cluster 2. Cluster 2 mainly represents the pollution caused by the accumulation of PM$_{2.5}$ concentration by local source
emissions. In addition, the contribution of secondary nitrate to PM$_{2.5}$ in cluster 2 is much larger than that of cluster 1. The
contribution of secondary source to PM$_{2.5}$ concentration is as high as 67.8%, and the secondary conversion of local source
emission is more obvious. In clusters 1 and 2, SOA is a source of high contribution to total PM$_{2.5}$ concentration. Both
regional and local source emissions contain a large amount of organic matter, which is an important component of
secondary source pollution. The dust source is also an important source of pollution caused by local emissions leading to
atmospheric PM$_{2.5}$ pollution.

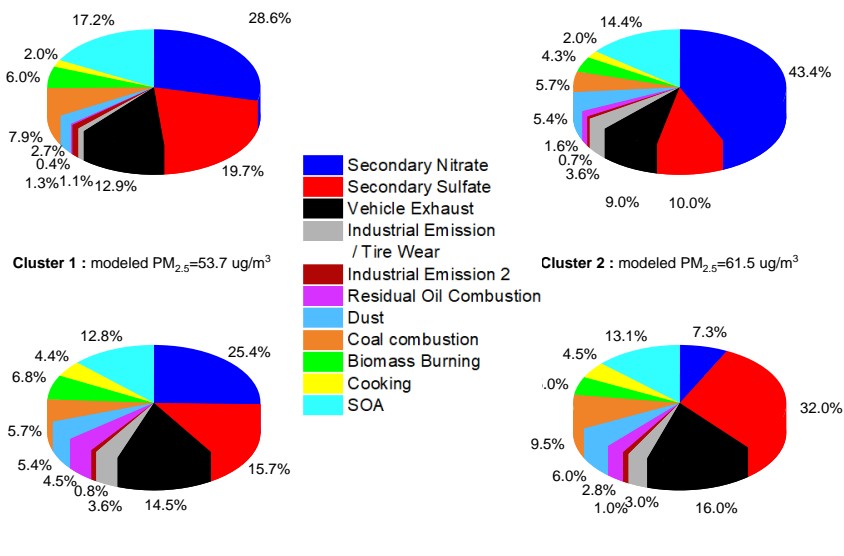


**Figure 7. The percentage of sources under different air mass clusters.**

Under cluster 3 and 4, PM$_{2.5}$ concentrations are relatively lower. Their biggest difference is the contribution of

secondary sulfate and secondary nitrate in cluster 3 and cluster 4. Under the influence of the northern air mass transport,
the proportion of secondary sulfate in cluster 4 is much larger than that of secondary nitrate, and also greater than the
proportion of secondary sulfate in cluster 3. It indicates that cluster 4 is discharged from coal combustion in winter in
northern China. And cluster 4 is a long-distance air mass transport, nitrogen oxides are more active in the atmosphere, and
secondary nitrate contributes less in cluster 4 than in cluster 3. In general, in the winter, the coal combustion in the north
of China and the biomass burning will affect the increase of PM$_{2.5}$ concentration in Shanghai under the action of air mass



movement. The accumulation of pollutants caused by local source emissions in Shanghai is also an important cause of
$PM_{2.5}$ pollution in winter.

**3.1.3. Source contribution under different episodes**

In order to better understand the source composition characteristics of atmospheric particulates during pollution
($PM_{2.5}>75\mu g/m^3$), we selected three time periods for further analysis, $PM_{2.5}$ concentration in the three periods is over
$75\mu g/m^3$ continuously or most of the time, as shown in Fig.8 (shaded area). The first episode (EP1) took place from 9 a.m.
to 9 a.m. on November 19, 2018, and lasted 24 hours. The second episode (EP2) took place between 19 p.m. on November
24 and 1 p.m. on November 26, 2018, during which the $PM_{2.5}$ concentration decreased, but soon began to rebound. The
third episode (EP3) lasts for a long time period, from 13 p.m. on November 27, 2018 to 7 a.m. on November 30, 2018, and
the $PM_{2.5}$ average concentration was the highest among the three episodes. It is worth noting that the pollution episode that
occurred from November 9 to 12 was not selected because the concentration of $PM_{2.5}$ was only over $75\mu g/m^3$ for a few
hours.

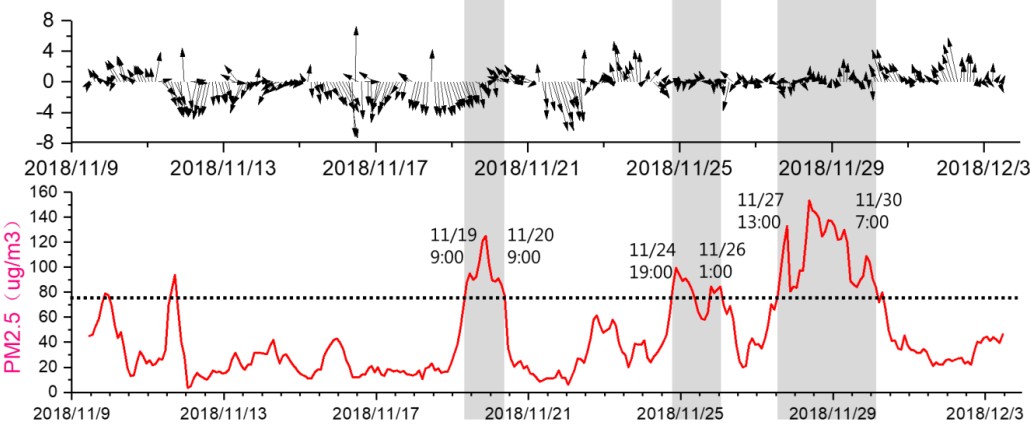


**Figure 8. Pollution episodes and its $PM_{2.5}$ concentration level**
The chemical characteristics of the selected three episodes are shown in Figure S5. It can be seen that the percentage
of each component of $PM_{2.5}$ in the total $PM_{2.5}$ under three episodes has little difference, indicating that the component will
not change much in a short time period. By analyzing the sources of $PM_{2.5}$ in the three selected periods, it is found that the
sources of $PM_{2.5}$ do not follow this rule, as shown in Fig.9. To better understand the influence of regional sources on factors
analyzed by PMF model, using the NOAA HYSPLIT model (https://ready.arl) for the three episodes, the 36-hour duration
backward trajectory of the 100-meter AGL was calculated. 0.5 global data assimilation system meteorological data was
deployed every 6 hours.

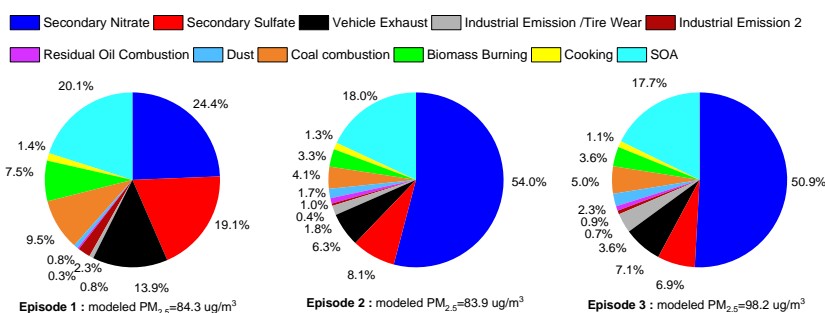

Figure 9. Source contributions under different episodes.

Combining the trajectories of the three episodes (EP1: a; EP2: b; EP3: c& d, Fig.10), the process from pollution
occurrence to pollution dissipation is a process that centers around the monitoring site, and the air mass entering the
monitoring point from the north turns clockwise from the east to the south and enters the monitoring area. And with the
increase of air mass movement in the vertical direction.
The average PM$_{2.5}$ concentrations observed during the three episodes were EP1: 96.2 μg/m$^3$, EP2: 79.8μg/m$^3$, and
EP3: 109.1μg/m$^3$, EP3>EP1>EP2. Generally speaking, PM$_{2.5}$ concentrations are highest when the airflow originates from
the mainland under prevailing northerly winds. When easterly and southerly winds prevail, the PM$_{2.5}$ concentration
observed under the influence of oceanic air mass is lower than that of air mass originating from the north. EP3 is greatly
affected by the transition air mass, and the diffusion of adverse pollutants caused by the weak air flow in the vertical
direction and small local wind speed leads to the accumulation of pollutants, resulting in the highest concentration of PM$_{2.5}$.
EP2 is mainly affected by the clean air flow from the south and some ocean air masses, and the vertical movement of the
existing air flow is greater than EP3. Compared with EP2 and EP3, EP1 has a greater contribution to secondary sulphate
factor under the influence of the northern continental air mass, which was close to the study of Hua et al. (2018) on the
source analysis of PM$_{2.5}$ in Beijing area. Additionally, secondary sulfate factor may also be affected by primary emission
of coal combustion. Under the influence of the air quality of marine air mass and transition air mass, EP2 and EP3 show
considerable contribution of secondary nitrates (EP2: 54.0% and EP3: 50.7%), while when the land air mass is dominant,
the contribution rate is much lower (EP1: 24.1%). The contribution of biomass burning and coal combustion source factors
show a changing pattern similar to that of secondary sulfate factors, the average contribution is the highest under the
influence of northern air mass, followed by transitional and marine air mass. Under the influence of the southern air mass,
the contribution of biomass burning and coal combustion source factors is the smallest, while the dust source is more





affected by local and short-distance air mass transport.

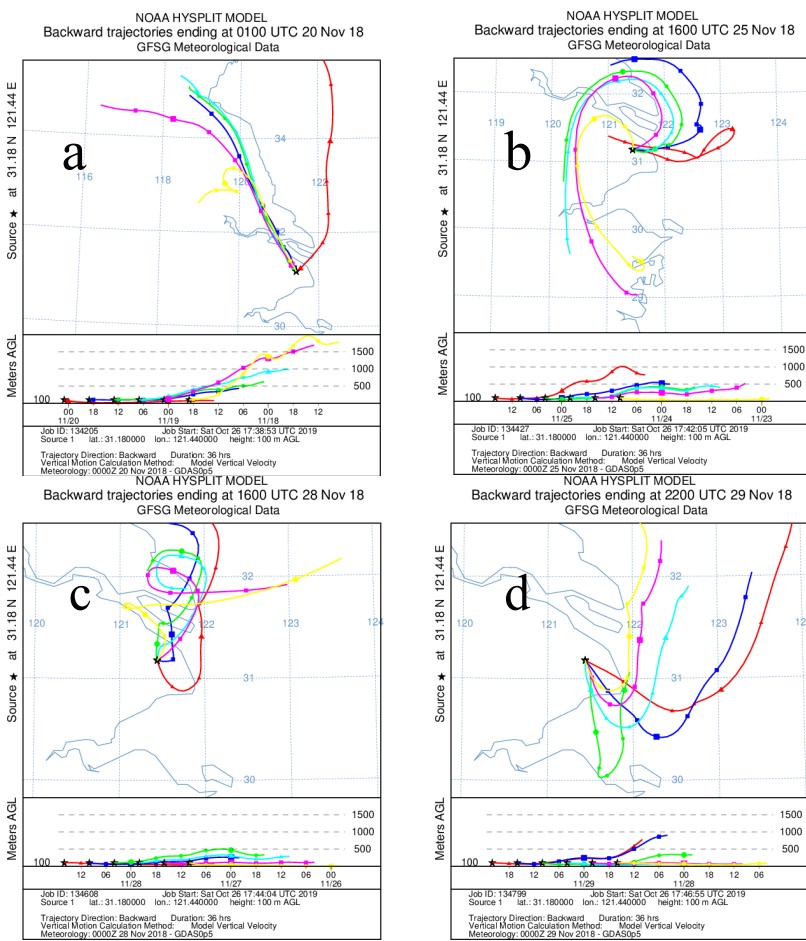


**Figure 10. The backward trajectory by each episode.**

In EP1, SOA is the largest contributor to the combined effects of anthropogenic and biogenic sources (19.8%), mainly

affected by the secondary conversion of OM carried by long-distance air mass from the north, while vehicle exhaust is
mainly affected by the continental air mass, the secondary conversion is not obvious, and other factors have little effect. In
general, due to differences in energy structure and production and living habits, biomass burning and coal combustion
sources are greatly affected by airflow from the north, which indirectly affects the proportion of secondary sulfate and
secondary nitrate in $PM_{2.5}$. Air mass from the south and ocean is cleaner than air mass from the north. $PM_{2.5}$ pollution is
also affected by the vertical movement of air mass and horizontal wind speed.


**3.2 Impact of organic tracers on source apportionment**
We also tested the PMF model without including the organic tracers to compare with the base PMF (MM-PMF), which
includes the organic tracers. The input data for PMFt are given in Table S6. In PMFt, eight factors were resolved (see
source profiles in Figure S6), and the three factors, biomass burning, cooking, and SOA could not be extracted due to the
lack of the corresponding organic tracers. The correlation of the common factor contributions for each factor between PMFt
and MM-PMF is shown in Table 3. Generally, the eight common factors, except for secondary sulfate and vehicle exhaust,
the other six factors correlated well between the two runs (R = 0.906-0.993), indicating the robustness of the resolved
factors. The secondary sulfate adds more organic matter (PAHs, organic acids, etc.) to the MM-PMF factor profile, showed
a moderate correlation (R=0.665) between PMFt and MM-PMF.
**Table 3. Correlation (R) of common source factors between PMFt and MM-PMF**

| MM-PMF \ PMFt | Secondary Nitrate | Secondary Sulfate | Vehicle Exhaust | Industrial Emission /Tire Wear | Industrial Emission 2 | Residual Oil Combustion | Dust | Coal Combustion |
|---|---|---|---|---|---|---|---|---|
| Secondary Nitrate | **0.906** | 0.256 | 0.674 | 0.388 | 0.407 | -0.036 | -0.086 | 0.458 |
| Secondary Sulfate | 0.187 | **0.665** | -0.011 | -0.311 | 0.387 | -0.337 | -0.377 | 0.29 |
| Vehicle Exhaust | 0.352 | -0.265 | **0.562** | 0.315 | 0.321 | -0.063 | 0.094 | 0.514 |
| Industry / Tire Wear | 0.413 | -0.225 | 0.412 | **0.991** | 0.151 | 0.072 | 0.346 | 0.236 |
| Industry 2 | 0.523 | -0.033 | 0.289 | 0.061 | **0.983** | -0.144 | -0.17 | 0.582 |
| Residual Oil Combustion | -0.064 | -0.346 | -0.072 | 0.069 | -0.167 | **0.993** | 0.264 | -0.115 |
| Dust | -0.169 | -0.376 | -0.196 | 0.277 | -0.181 | 0.223 | **0.98** | -0.026 |
| Coal combustion | 0.573 | -0.027 | 0.436 | 0.153 | 0.636 | -0.122 | -0.013 | **0.967** |

The factor profiles of PMFt of the 8-factor solution is shown in Figure S6, and the difference of individual factor
contribution to PM$_{2.5}$ and to OC from MM-PMF and PMFt are show in Fig.11. And, among MM-PMF and PMFt results,
the concentrations of reconstructed PM$_{2.5}$ and OC are shown in Table S7.Compare the factor profiles and contributions of
MM-PMF and PMFt, it can be found that the contribution from the secondary sources changed from (29.1 μg/m$^3$) 63.8%
of MM-PMF to (28.5 μg/m$^3$) 63.9% of PMFt, while SOC changed from (2.9 μg/m$^3$) 46.1% to (3.1 μg/m$^3$) 48.9% of the
total OC mass. The change in the factor profile of the secondary nitrate to PM$_{2.5}$ is not obvious, and the contribution of
secondary nitrate to PM$_{2.5}$ was greater than that of secondary sulfate. While for OC, the contribution of secondary sulfate
was greater than that of secondary nitrate in the two PMF analysis results. In addition, the source apportionment results
under two scenarios showed that secondary sources accounted for 63.9% and 63.8% of the total mass of PM$_{2.5}$ in PMFt
and MM-PMF, respectively.





With the addition of organic tracers, contribuiton from vehicle exhaust dropped from 14.5% to 11.3% of MM-PMF,
the correlation (R) of the vehicle exhaust between MM-PMF and PMFt was only 0.562 (Table 3), which is mainly because
organic matters were added to the vehicle exhaust factor of MM-PMF, and the factor contribution time series diagram has
changed. There was no significant change in the contribution of vehicle exhaust to OC.
The contribution of industrial emission 2 and residual oil combustion sources to PM$_{2.5}$ mass changed from 2.0% and
3.1% in PMFt to 1.0% and 1.8% in MM-PMF. This is mainly because OC shifted from these two factors to the newly added
source of MM-PMF. Due to the transfer of OC and other elements (Ca, and Zn, etc.) in the coal combustion, the proportion
of this factor in PM$_{2.5}$ decreased from 8.5% of PMFt to 6.7% of MM-PMF. Compared with MM-PMF, the concentration
of OC in industrial emission 2, residual oil combustion and coal combustion in PMFt increased by 0.16, 0.14 and 0.25
μg/m$^3$, respectively.Generally, for the eight factors, the factor industrial emission /tire wear and dust were more stable,
contributions of these two factors to total PM$_{2.5}$ were relatively small, with little differences between MM-PMF and PMFt
results (3.0% vs 2.9% and 5.0% vs 5.1%), whereas the contribution from dust to the total OC mass changed from (0.10
μg/m$^3$) 1.1% of MM-PMF to (0.24 μg/m$^3$) 3.8% of PMFt. Compare the difference of individual factor contribution to OC
from MM-PMF and PMFt, POC changed from 53.9% to 51.1% of PMFt of the total OC mass.

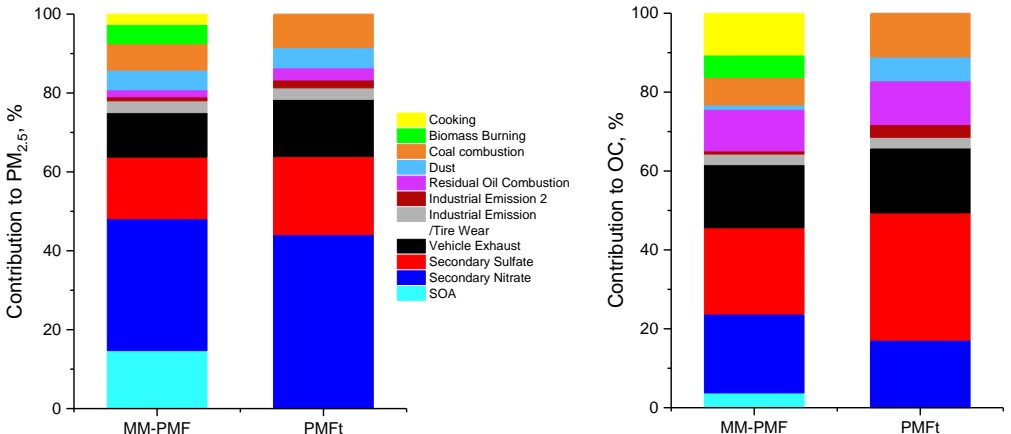


**Figure 11. Difference of individual factor contribution to PM2.5 and to OC from MM-PMF and PMFt.**
**4. Conclusions**
In this study, an intensive observation campaign was organized in winter to gain more insights into the sources and



formation of PM$_{2.5}$ in Shanghai, a typical city in the Yangtze River Delta region. PM$_{2.5}$ and its chemical components,
including water-soluble inorganic ions, carbonaceous species, trace elements and organic markers (PAHs, sugars, organic
acids, etc.) were measured with 1-h time resolution. By combining comprehensive data sets of chemical species into the
PMF model for source analysis, the average contribution of secondary pollution sources is more than 60%. The MM-PMF
with organic tracers added was further compared with the traditional PMF without organic tracers. MM-PMF further
resolved the source contributions from SOA, biomass burning and cooking factors, which can't be separated without
organic tracers.

Comparing the contributions of different sources to OC mass from MM-PMF, it can be seen that SOC and POC

contributed 45.1% and 54.9%, respectively. The SOC associated with secondary nitrate and sulfate factors accounted for
2.68 μg/m3 (41.3%) on average across all the study. The source of vehicle exhaust contributes the most to POC, in addition,
it is worth noting that the contribution of cooking to PM2.5 mass only account for 2.6%, but it contribute 10.7% to OC.

Comparisons of PM$_{2.5}$ source composition under different air quality shows that the secondary nitrate contribution is

much higher when PM$_{2.5}$ concentrations are high. The data during the whole observation period were analyzed by backward
trajectory clustering and four types of backward trajectories were analyzed separately. It was found that secondary nitrate
was the main cause of air pollution in Shanghai. In addition, in the absence of pollution, vehicle exhaust sources still make
a significant contribution. In winter, Shanghai area is greatly affected by the air mass from the northern area, which is an
important cause of particulate pollution. In addition, adverse meteorological conditions may also cause the accumulation
of particulate matter, resulting in air pollution.

*Competing interest.* The authors declare that they have no conflict of interest.

*Acknowledgement.* This study was financially supported by the National Natural Science Foundation of China (NO.
41875161) and Hong Kong Research Grants Council (16305418). We thank Shanghai Academy of Environmental Sciences
for logistic help with the TAG-GC/MS measurements.

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
