# Peer review of "Source apportionment of PM2.5 in Shanghai based on hourly organic molecular markers and other source tracers"

_Atmospheric Chemistry and Physics, 2019_

## Referee Comment (RC1) · Anonymous Referee #1 · 16 Apr 2020

General comments:

This paper presented source apportionment and chemical characteristics of fine aerosols in Shanghai, China. This work was performed based on highly time-resolved measurements during a near one-month campaign in winter, along with PMF analysis. The combination of different aerosol compositions, including OC, EC, water-soluble ions, organic markers, and some other tracer elements was applied to constrain the PMF analysis. I agree such a methodology could help to further understand PM2.5 sources and its processes in the atmosphere. However, I don't support that the data collected from the two different sites could be reasonably combined together into the

[Figure]

PMF analysis. This data treatment strategy can lead to a huge uncertainty in PMF results. Therefore, I don't suggest this paper can be accepted, if the authors wouldn't fully address the uncertainty generated from the two databases in PMF analysis. When carefully reading this paper, one can find it's not well written, and it's hard to follow. Moreover, some discussions could not be directly supported by the results. It should be suggested that all (or major) authors should carefully review/validate this manuscript before submitting.

Comments in detail:

Lines 14-21: Abstract should be concise. These sentences can be improved to only briefly describe what necessary and importance of this study are, as well as to briefly address scientific question of this study.

Line 19: Measurements at the two different sampling sites were conducted in this work, which should be addressed here. This is a very important information in PMF analysis. Aerosol chemical composition, especially organics, can be substantially different at different sites even in a same city region. Again, I don't believe this methodology is reasonable unless the authors can prove it.

Lines 21-22: This sentence can be improved. And what's the difference between "molecular makers" and "organic makers" here? Please rewrite this sentence.

Lines 23-24: Not clear. Please rewrite it.

Line 24: "The three" -> "These three". Another question, why did the authors select those sources for the quantification discussion here? Why not others included, since you have identified many source factors?

Line 25: It has already been well known about "the secondary source is an important source of atmospheric pollution" in eastern China. Better to rewrite this sentence.

Lines 26-27: It's a confusing sentence (Grouping different sources . . ., respectively). Please rewrite it. And what is the difference between SOA and SOC mentioned here

together? Maybe only needing to show any individual of the two terms in the abstract section, if you don't need to explain the different things between SOA and SOC. In addition, SOC and POC should be defined before using them.

Lines 33-35: Maybe a region scale should be given here, for instance, e.g., in eastern Asia or China?

Lines 35-36: This sentence should be revised to be better read. In addition, "Zhang et al., 2009;)." -> "Zhang et al., 2009)."

Lines 36 – 42: These sentences should be revised. As written here, aerosol climate and environmental impacts are not well described. They can be improved better for easy follow. For example, the sentence (lines 38-40) is discussing about "climate impact of aerosols", while in the hereafter sentence, you conclude "importance for air quality". I mean, in this paper, it may be not needed to highlight aerosol climate aspects in too much detail, but it can be discussed with a very brief version. In addition, in the first sentence (lines 33-35), you already have addressed the situation of PM2.5 pollution and it's important. But, in the sentences (lines 40-42), you highlighted the similar information again. Overall, I would suggest that more concise sentences can be made here to better describe importance and necessity that are associated with the work of the present study.

Lines 43-44: Again, this information has been declared in the above paragraph, no? Is the sentence of lines 44-45 expressing a similar knowledge as the lines 35-36?

Lines 46-47: What are ". . . other methods"? They could be briefly discussed here, since you would highlight some advantages of PMF model, especially you will need apply it with a unique PMF input dataset.

Lines 48-52: It may be not only due to this reason (as you discussed above) why PMF has been widely applying in aerosol source apportionment, as reported by many previous studies. Meanwhile, it would not be necessary to list too many references

here. If I understood well, the authors would like to highlight the advantage of PMF compared to other receptor models, and why PMF was applied in the present study.

Line 55: Filter measurements can be also available for short-term source apportionment study. Not sure if this sentence is necessary/clear here with the current way.

Lines 58-62: These sentences are not clear enough. What are advantages of the observation methods in the present study, compared to AMS techniques? One could argue that the combination between AMS and organic tracers could be also a very good strategy for source apportionment of organic aerosols (e.g., Huang et al., 2014 Nature).

Lines 63-64. One transmission sentence between the two sentences is needed, otherwise one could understand that many previous studies performed in Shanghai could be only due to a large population. I guess you would express that Shanghai is a megacity, representing for a typical economic zone in China and with a large population there, while air pollution issues are complex due to unclear enough aerosol sources that have to be better understood to further improve air quality.

Lines 64-71: Following the similar comments above, I would suggest the author to improve this paragraph.

Lines 67-68: It is not true that offline techniques are limited in air pollution event studies.

Line 77. As I mentioned above, the reasonable application to combine the two data sets from the two different sampling sites for PMF analysis needs to be further proved.

Line 95. More detail in methods of organic compounds data analysis (including more descriptions in instrumental information) and data quality control should be described here, since such data set is a major part in the present study.

Line 131: "1/2" can be understood by "1 or 2" or "0.5". Same comment on line 132: "5/6".

Line 134. A refence or acknowledgement when you use trajectory data and tool should be given.

Lines 142-143. Some references should be given to support this statement. And what the link between this sentence and the ones hereafter? It seems that further discussion on potential impacts of the meteorological conditions on PM2.5 pollution observed in the present study is needed. Wind-dependent analysis of different aerosol sources can be applied here, I think.

Line 167: Define the terms BS, DISP and BS-DISP.

Line 169: Please show specific results and/or further prove the cases when the factors were mixed based on those base runs.

Lines 172-173: Figure 3 presents results resolved by the MM-PMF analysis. This information should be given in this sentence, since two PMF methods have been applied and hereafter you did a lot of discussion on Figure 3.

Line 173: "Secondary nitrate factor (F1) was identified by high concentrations of nitrate and ammonium...". The PMF factor profile is described by contributions of each variable, but not their absolute concentrations. I think it might be enough when you directly discuss their contributions in factor profiles. Same comment on the similar statements.

Lines 177-179: what's difference between SOA and SOC? Please show results about "... which can be seen from the correlation of the respective species with OC.".

Lines 179-181: Those sentences could be read by: "Figure 4 presents the relative contributions of various source factors to PM2.5 and OC based on MM-PMF analysis, respectively. The F1 and F2 contributed 33.5% and 15.6% to the total PM2.5 mass concentration, respectively."

Lines 181-182: What does SOC contribute to, PM2.5 or OC?

Lines 182-183: Such sentences would be better discussed step-by-step for each factor,

something like: "As shown in Fig. 5, the diurnal variation of the secondary nitrate factor shows high concentration during nighttime (e.g., 21:00 – 04:00), while its low concentration is observed during daytime (e.g., 11:00-14:00). etc. The F1, a secondary sulfate-rich factor, didn't present obvious diurnal pattern, which is however characterized by limited variations across the day. etc...." This is only an example. The authors don't need to exactly follow it, but clearer discussions are needed. In addition, influence of air temperature on the nitrate-rich factor should not be ignored, due to volatile properties of particulate ammonium nitrate.

Line 186: please show evidence to further demonstrate influence of regional transportation on the F1, when you addressed it here.

Lines 187-189: the diurnal variations of RH and temperature can be plotted to support "high RH condition during the day". But it's not convinced to address that no obvious diurnal variation of sulfate-rich factor could be mainly due to the reason here. Other reasons should be further discussed. In addition, the conclusion of the sentence (lines 188-189) is weak, which is not sufficiently supported by the results.

Lines 192-194: This sentence could be modified by "The factor 3 (F3) profile presents a high abundance of EC, OC, Ca, and Cu, as well as it contains some organic tracers (PAHs and organic acids, etc.), ...". Lines 192-194: In addition, what does contribute to 11.3% of the total PM2.5 mass on average? An information to be needed to transfer this sentence to the next, to indicate this factor could be linked to vehicle exhaust. Otherwise it's hard to understand if this factor 3 is associated with vehicle exhaust source when you start to discuss it in the next sentence. What does this "accounting for 16.2% to OC" refer to, the total vehicle exhaust emissions or only POC?

Lines 194-196: This sentence is not clear. Please revise it.

Line 197: what is "daily variation"?

Lines 197-198: In addition, "xxx" has high correlation with NOx and CO, respectively,

...? Mistakes in "R=67" and "R=58".

Lines 198-201: Needs to revise these sentences for better understanding.

Line 204: "loads" -> "loadings. Were you discussing on profile in this sentence? Please revise this sentence.

Line 207: The "F4" needs to be defined.

Lines 207-209: I think this sentence can be better improved.

Line 210: a mistake in "R=49".

Lines 209 – 210: I don't understand why you discuss on "F2" in the first phase of this sentence, then you get conclusion about "F4"?

Line 210: "source.The" -> "source. The". What does "the factor" refer to?

Lines 210-213: More discussions to be needed to further support "suggesting the benefit of the online high-resolution measurement.".

Line 213: "measurement.The" -> "measurement. The"

Lines 213-214: "are often used in industrial processes" -> "are often used as industrial emission markers". But I am not sure if you did this meaning? In addition, the figure that you are discussing should be pointed out.

Line 215: a mistake in "R=59".

Line 215: what did you discuss in this sentence "No diurnal variation was observed."? This is not clear at all.

Line 215: "The factor" refer to the factor 5?

Lines 216-217: This sentence can be improved for better reading.

Lines 217-218: how could you prove your factor 6 that mainly comes from ship transportation? Further discussion/evidence to be needed.

Line 218: again, this sentence is not clear: "The diurnal contribution at night was greater than that during daytime.". Please the authors carefully check such similar issues over the entire manuscript.

Lines 219-220: to me "the contribution (10.7%)" is relatively high, but I don't think "is not negligible" is an appropriate use here.

Lines 228-229: Mistakes in "R=59" and "R=56".

Line 230: "such as PAHs present in the source" -> "such as PAHs presented in the source profile"? or a figure you discussed should be linked here. "The results are different" -> "These results are different"

Lines 232 – 233: the two factors should be discussed separately.

Lines 234 – 236: literatures should be cited here to support this statement. In addition, "K" needs to be defined.

Line 236: "This factor, contributed 4.9% and 5.7% of total PM2.5 and OC mass on average, respectively." -> "On average, this factor contributed 4.9% and 5.7% to the total PM2.5 and OC mass concentrations, respectively."

Line 237: "biomass burning contains high-loading" -> "the source profile of biomass burning factor contains high loadings . . .(Fig. ?)".

Line 238: "(including coal combustion, biomass burning, etc.)" -> "(including coal combustion and biomass burning, etc.)".

Lines 239-241: To be suggested to discuss the two factors, biomass burning and cooking, separately.

Lines 242 – 243: which figure/table did you discuss in this sentence?

Line 243: This value "14.7%" is not a major fraction, but it can be called as a relatively high fraction.

Lines 244 – 245: Do the authors mean that this SOA factor could be associated with a mixture SOA factor from anthropogenic and biogenic sources? One could argue very small fraction of biogenic emission during wintertime in Shanghai region. In addition, it is not sufficient by the similar diurnal variations between nitrate and SOA to indicate that they have similar formation processes and interaction between anthropogenic-biogenic sources. Further discussion is needed to clearly explain such a factor and its potential formation processes.

Lines 258 – 264: those sentences should be discussed associated with your results when you need them to support something. Or if only for a summary purpose here, those contents could be moved into the introduction section.

Line 270: "It can be seen from Fig.7 that the composition varies with the source of the air mass." -> "As shown in Fig. 7, the PM2.5 sources vary evidently associated with different air mass origins."

Lines 273-275: those sentences can be improved to be better read.

Line 276: cluster 1 is not associated with North China, but it could be more reasonable coming from eastern China.

Lines 283 – 284: could be possible regional transport contributing to dust observed at this receptor site? Wind analysis together with dust source could help to further investigate this point.

Lines 287 – 292: the clusters 3 and 4 could be associated with relatively clean air masses originated from ocean areas (see Fig. 6). The differences between clusters 3 and 4 are a long-range and a short-range transportation characterization, respectively. More locally-formed pollutants could be expected observed within cluster 3 air masses, while more regional pollutants could be linked to cluster 4.

Lines 297-299: Please rewrite this sentence.

Lines 299-300: Please revise this sentence for better reading.

Line 312: why the authors used the different HYSPLIT model versions in this paper? It should be declared in the main text as well.

Lines 330-331: do the authors have any evidence to support this statement: " secondary sulfate factor may also be affected by primary emission of coal combustion."?

Lines 331-333: this sentence can be better improved.

Lines 333-335: give a link (e.g., figure or table) to your results what you are discussing.

Lines 335-336: a same comment as the above, give a link. In addition, which air mass trajectories are shorter, to support your statement "the dust source is more affected by local and short-distance air mass transport."? It can be additionally/more directly helped to check wind dependence of your source factors (especially for local factors), in addition to air mass analysis.

Lines 342-335: very weak evidence from the present study to support this sentence.

Lines 345-346: again, do you have evidence to support this? If yes, please show that.

Line 347: I guess this section might be better moved to methodology section or at beginning of section 3.1. Moreover, I guess it would be also worthful for doing correlations (with R and slope values) of individual component (e.g., nitrate, sulfate, ammonium, POC, SOC, EC, Ca2+, and K+) between the two different PMF analysis.

Lines 357-358: This sentence can be divided into two sentences to address the two different things. In addition, line 357 "is shown" -> "are shown"; line 358 "are show" -> "are shown".

Line 359: "Table S7.Compare" – "Table S7. Compare"

Lines 384-402: I think this conclusion section needs a major revision to highlight the key findings and further atmospheric implications from the present study, in addition to some technical corrections what I have listed below:

Line 385: "a typical city" -> "a megacity"

Line 388: what does "contribution of secondary pollution sources" contribute to?

Line 392: Please rewrite this sentence.

Lines 400-402: Very common statements, since they are well-known knowledges and have been reported by many previous studies. As I commented above, the authors would be able to better introduce what the key findings/implications supported by the present study.

Comments on figures:

Figure 1 should be improved by adding a larger scale map to indicate where Shanghai is located in China or eastern Asia.

Figure 2 and Figure 8 should be merged together by a single figure. There is no need to repeat PM2.5 temporal variations again in the main text. If you wanted to highlight its relation with wind direction, you can add wind direction in Figure 2.

In figure 3, I think it's better to draw the time series of those sources with lines than using solid-filled areas, since we cannot see clearly if there is a missing gap for the sampling.

Figure 4: Again, without proving your reasonable PMF analysis using the data from different sites, it isn't convinced to accept these quantitative results.

Figure 5: why did the authors show two-hour resolution diurnal variations instead of using hourly, since you have hourly data as mentioned in previous section? I suggest to use hourly.

Figure 6 and Figure 7: I believe that it can be more visualized to merge the two figures within a single one, e.g., different clusters associated with different pie charts.

Figure 9: a same as Figure 4.

Figure 11: The mass concentrations of those source factors can be also presented here to better understand the differences between the different PMF analysis methodologies.

---

## Referee Comment (RC2) · Anonymous Referee #4 · 24 Apr 2020

Li et al. identified sources of air pollution in Shanghai and quantified their contributions using the PMF model. This study fits the scope of ACP and has some interesting findings. The paper is well structured, while there are many grammar mistakes that needs to be fixed and many sentences that needs to be revised to improve clarity. I also have questions about its methodologies.

Major comments:

1. The authors would need to explain why the data from the two sites can be combined and how this might lead to uncertainties in the results.
2. It is not clear to me how PM2.5 concentrations for the four clusters of air mass were calculated. Please elaborate.
3. Misuses of words and sentence structures sometimes can make it hard for readers to understand. I just listed below some as examples, but there are more mistakes that need to be fixed.

Minor comments:

"While" and "however" are misused many times: For example, in Line 241

28-29: This sentence is grammatically incorrect.

77: What does "change of sources" mean? Change relative to what?

159: Please split this sentence into two sentences.

195: "accounting for 16.2% to OC" -> "and accounts for 16.2% of OC"

199: 67->0.67, 58->0.58

211: 49->0.49

229: There is a grammar mistake. Please split this sentence into two sentences.

284: I'm not sure how the authors concluded that dust are local emissions, since the dust contribution in Cluster 2 is actually lower than the contribution to Cluster 1. Also the sentence is redundant, and could be revised to "Dust is also an important local emission source of PM2.5."

401 - 402: Please revise the sentence to something like "Air quality in Shanghai area is greatly affected by the air pollution transport from the northern regions. "

Figure 8: The upper panel needs y axis label

---

## Author Comment (AC1) · 20 Jul 2020

**Point-by-point response to comments by Reviewer#4**

We thank the reviewer for the detailed and constructive review comments. Below is our point-by-point response to each comment, marked in blue. Changes made to the main text are also marked in blue in the revised manuscript file.

**General comments:**

1. The authors would need to explain why the data from the two sites can be combined and how this might lead to uncertainties in the results.

**Response:** This comment has been addressed in the point-by-point response to comments by reviewer#1, in **"Response to General comments".** The reviewer can refer to the corresponding response there.

2. It is not clear to me how PM2.5 concentrations for the four clusters of air mass were calculated. Please elaborate.

**Response:** 36-h duration backward trajectories arriving at an altitude of 100 m above ground level over the sampling site were calculated for each hourly sample, deploying the 0.5 Global Data Assimilation System meteorological data. Thus, for 289 samples, there are 289 trajectories. The trajectories were then classified into four clusters according to the geographical origins and movement process of the trajectories using the TrajStat model. Each sample was then attributed to its corresponding cluster from the model. The average $PM_{2.5}$ concentrations for each cluster are calculated by averaging all the $PM_{2.5}$ concentrations from the samples belong to that cluster.

3. Misuses of words and sentence structures sometimes can make it hard for readers to understand. I just listed below some as examples, but there are more mistakes that need to be fixed."

**Response:** we sincerely thank the reviewer to give us a chance to revise our manuscript. We have carefully revised the whole manuscript with substantial efforts. Please see the detailed changes made in the updated version of the manuscript.

**Minor comments:**

"While" and "however" are misused many times: For example, in Line 241

**Response:** revised accordingly. We have carefully examined the whole manuscript and improved the language.

28-29: This sentence is grammatically incorrect.

**Response:** The sentence has been rewritten.

**Lines 25-26:** "Cooking emission was a minor contributor (2.8%) to $PM_{2.5}$ mass while a significant contributor (11.4%) to the OC mass."

77: What does "change of sources" mean? Change relative to what?

**Response:** it was a typo mistake and caused ambiguity. This sentence has been rewritten in the main text.

**Lines 76-77:** "The results from this work can provide support for the development of air pollution prevention and control strategies."

159: Please split this sentence into two sentences.

**Response:** revised as suggested.

**Lines 181-182:** "The preferential input species for PMF analysis are those with high abundance and known to be specific to certain sources. Generally, organic markers with lower volatility and lower reactivity were selected as input species for MM-PMF."

195: "accounting for 16.2% to OC" -> "and accounts for 16.2% of OC"

**Response:** revised as suggested. These sentences have been rewritten in the main text and updated with the data of the model results.

**Line 227:** "F3 contributed to 12.6% of the total $PM_{2.5}$ mass and 19.4% of OC on average."

199: 67->0.67, 58->0.58

**Response:** revised as suggested. This sentence has been rewritten and updated with the data of the model results.

**Lines 226-227:** "In addition, F3 has high correlations with $NO_x$ (R=0.68) and CO (R=0.48), further supporting the association of this factor with vehicle exhaust."

211: 49->0.49

**Response:** revised as suggested.

**Lines 232-233:** "F4 shows a high correlation with $NO_x$ (R=0.49), and $NO_x$ in the Yangtze River Delta mainly originates from industrial and vehicular pollution sources (Fu et al.,2013)."

229: There is a grammar mistake. Please split this sentence into two sentences.

**Response:** revised as suggested.

**Line 257:** "F8 contains a high abundance of As and Pb, which identifies this factor ro be associated with coal combustion (Chen et al., 2013)."

**Line 261:** "F8 contributes to 5.3% of total $PM_{2.5}$ and 5.6% of OC, respectively."

284: I'm not sure how the authors concluded that dust are local emissions, since the dust contribution in Cluster 2 is actually lower than the contribution to Cluster 1. Also the sentence is redundant, and could be revised to "Dust is also an important local emission source of $PM_{2.5}$."

**Response:** this weak statement has been removed from the main text.

401 - 402: Please revise the sentence to something like "Air quality in Shanghai area is greatly affected by the air pollution transport from the northern regions."

**Response:** the whole conclusion section has been revised to make it more concise.

**Lines 415-417:** "The results indicated that PM pollution in winter in Shanghai area is greatly affected by both local pollutant emissions and the regional transport from the northeastern continental regions."

Figure 8: The upper panel needs y axis label

**Response:** suggestion taken. Original Figure 8 was combined with Figure 2, as suggested by one reviewer. See the revised version in **Figure 2**.

---

## Author Comment (AC2) · 20 Jul 2020

**Point-by-point response to comments by Reviewer#1**

We thank the reviewer for the detailed and constructive review comments. Below is our point-by-point response to each comment, marked in blue. Changes made to the main text are also marked in blue in the revised manuscript file.

**General comments:**

This paper presented source apportionment and chemical characteristics of fine aerosols in Shanghai, China. This work was performed based on highly time-resolved measurements during a near one-month campaign in winter, along with PMF analysis. The combination of different aerosol compositions, including OC, EC, water-soluble ions, organic markers, and some other tracer elements was applied to constrain the PMF analysis. I agree such a methodology could help to further understand $PM_{2.5}$ sources and its processes in the atmosphere. However, I don t support that the data collected from the two different sites could be reasonably combined together into the PMF analysis. This data treatment strategy can lead to a huge uncertainty in PMF results. Therefore, I don t suggest this paper can be accepted, if the authors wouldn't fully address the uncertainty generated from the two databases in PMF analysis. When carefully reading this paper, one can find it's not well written, and it's hard to follow. Moreover, some discussions could not be directly supported by the results. It should be suggested that all (or major) authors should carefully review/validate this manuscript before submitting.

**Response to General comments:** Thanks for raising this issue of combining data from two sites to do PMF analysis. It is our oversight not to have addressed this issue in our ACPD submission. We'll detail below the rationality of combining data from two sites and discuss associated uncertainty. We have also edited the entire manuscript text to improve the logic clarity.

**1. The rationality of combining data from two sites**

In this manuscript, we combined the $PM_{2.5}$ major component data from PD site and organic molecular markers data from SAES site for the source apportionment analysis. The two sites are 12 km apart, both are characterized as a general urban location. We agree that it would be desirable to use chemical composition data from one single site to carry out the source analysis. Unfortunately, in this work, the SAES site had measurements of organic tracers, major ions, and elements, but did not have ECOC measurements. The PD site had data of major ions, elements, and OCEC, but no organic tracer data. As a result, we had to resort to "borrowing" certain composition data from a similar site nearby. The data combination provided a more comprehensive dataset to fully characterize the $PM_{2.5}$ pollution sources, for general urban district in this region.

The detailed rationality of combining data from the two sites is explained in the following three sub-sections.

**1.1 The neighborhood characteristics of the two sites are similar.**

Shanghai's land area is part of the alluvial plain of the Yangtze Delta Region, with an average height of 2.19 meters above the sea. The average wind speed is around 3-4 m/s. The whole Shanghai area belongs to one air basin. The geographic distance between the two sites is 12.1 km. Figure S1 shows the surroundings of the two sites. Both sites are surrounded by large residential areas, in addition to scattered shopping malls and restaurants, indicating the similarities of mixed emissions influence for the two sites. The PD site is a designated urban monitoring station in Pudong district (Zhao et al., 2013a). The SAES site is located in Xuhui district and more pollution characteristics of this site can be found in Wang et al. (2018).

**1.2. The pollution characteristics at two sites are similar.**

$PM_{2.5}$ mass concentrations, gaseous pollutants (CO, $SO_2$, $NO_2$) and $PM_{2.5}$ carbonaceous components (i.e. BC, OC and EC) concentrations between the two sites are examined. Table S1 lists the campaign-average concentrations and percentage difference of the average concentrations between the two sites. Figure S2 compares the time series and correlations of

PM$_{2.5}$ mass, and gaseous pollutants (CO, SO$_2$, NO$_2$) during the measurement period between the two sites.

As shown in Figure S2, the pollution levels of PM$_{2.5}$ and NO$_2$ at two sites showed an excellent degree of agreement with each other, (R$^2$=0.92, slope=0.95) for PM$_{2.5}$ and (R$^2$=0.78, Slope=0.86) for NO$_2$. The site-site difference in their average concentration was less than 6% (Table S1). CO also showed a high correlation (R$^2$=0.78) and similar concentration levels at the two sites. SO$_2$ showed a moderate correlation between the two sites (R$^2$=0.45), its temporal variations were broadly in sync.

No OC and EC data were available at SAES site during the sampling period. Thus, we next compare related carbonaceous measurements. Specifically, BC measured by aethalometer at SEAS is compared with EC measured at PD (Figure S3), showing synchronous variations, and their concentration levels were also similar, with a percentage difference of 33%. AMS-measured PM$_{1.0}$ OA at SAES is compared with PM$_{2.5}$ OM mass estimated using OC at PD (Figure S3). The two quantities tracked very well between them throughout the measurement period, except for one OA episode lasting for a few hours on 29 Nov. Additionally, they showed comparable levels between two sites, with a relative difference of 8-17%.

**Table S1.** Concentrations and relative difference of PM$_{2.5}$ mass, carbonaceous components, and gaseous pollutants (CO, NO$_2$ SO$_2$) at PD and SAES sites during measurement period.

| Species | SAES site (µg/m$^3$) | | PD site (µg/m$^3$) | | Relative Difference |
|---|---|---|---|---|---|
| | avg | stdev | avg | stdev | |
| PM$_{2.5}$ | 47.5 | 34.3 | 46.3 | 33.8 | -2.5% |
| BC at SAES vs. EC at PD | 2.35 | 1.66 | 1.59 | 1.13 | -33.2% |
| OC | | | 6.48 | 2.79 | |
| AMS-measured PM$_1$ OA at | | | 11.66[a] | 5.09 | -17.4% |
| SAES vs. PM$_{2.5}$ OM at PD | 14.11 | 11.14 | | | |
| estimated using OC | | | 12.96[b] | 5.66 | -8.2% |
| Gaseous pollutants | | | | | |
| CO (mg/m$^3$) | 0.79 | 0.33 | 0.66 | 0.32 | -16.5% |
| NO$_2$ | 57.15 | 30.04 | 53.88 | 29.21 | -5.7% |
| SO$_2$ | 5.88 | 1.96 | 7.75 | 2.73 | 31.8% |

[a] Estimated PM$_{2.5}$ OM concentration assuming OM =1.8*OC

[b] Estimated PM$_{2.5}$ OM concentration assuming OM =2.0*OC

**1.3. PMF results using SAES data and using PD data are comparable.**

We have performed an additional PMF test with SAES-measured inorganic ions and elemental tracers as input (PMF$_{SAES}$: i.e. inorganic ions, elements and organic markers are all from SAES site). The OC and EC data are "borrowed" from the PD site as no such data are available from SAES site. The results of PMF$_{SAES}$ are compared with the PMF solution obtained using combined data from the two sites (MM-PMF in this work, alternatively, we will refer it as PMF$_{PD}$ for comparison here). PMF$_{SAES}$ showed very similar results compared with PMF$_{PD}$ in both apportioned source contributions (Figure R1) and in resolved source profiles (Figure R2). This comparison strongly suggests that due to the similar pollution characteristics at the two urban sites, it is feasible to "borrow" one site's major components and elements data to conduct the PMF analysis.

In summary, we have shown with measurement data that the major PM$_{2.5}$ components at the two urban sites are synchronous in temporal variations and highly similar in concentration levels. This provides data support to argue for the approach of using combined data set to achieve comprehensive source apportionment of PM$_{2.5}$ for the urban districts in this work. We interpret that the source apportionment results using the combined dataset are representative of the common and major

sources at PD and SAES sites. Otherwise it would not be possible to observe the highly similar level of $PM_{2.5}$. On the other hand, we recognize that the data combination approach would be ineffective to extract a potential source specific at one single site. More specifically, as pointed out by the reviewer, the differences of organic markers at the two sites may be non-negligible, thus, the apportioned results, especially the sources apportioned by certain organic markers (i.e., cooking in this study) could be site-specific to the SAES site alone. Nevertheless, the major source factors should be consistent at two sites and can be used to reflect the general urban pollution in our study location. Finally, a rigorous quantification of the uncertainties caused by the data combination at two sites needs side-by-side online measurements of PM compositions, especially organic markers at two sites, and currently this information is not available.

[Figure]

**Figure S1.** Surroundings of the two sampling sites (i.e. PD and SAES) in this study.

[Figure]

**Figure S2.** Comparison of PM2.5 and gaseous pollutants between PD and SAES sites during measurement period.

[Figure]

**Figure S3.** Comparison of PM$_{2.5}$ carbonaceous components between PD and SAES sites during measurement period. (OM/OC ratio from AMS measurement: 1.69±0.18 (1.4-2.0). 1.8→75[th] percentile of OM-to-OC ratio from AMS measurement; 2.0→ maximum value of OM-to-OC ratio from AMS measurement)

[Figure]

**Figure R1.** Comparison of factor contributions to PM$_{2.5}$ (a) and OC (b) between PMF combining two sites data (PMF$_{PD}$, i.e. MM-PMF in this work) and PMF with SAES data as input (PMF$_{SAES}$).

[Figure]

**Figure R2.** Comparison of factor profiles between PMF combining two sites data (PMF$_{PD}$, PMF with PD major components and elements data, i.e. MM-PMF in this work) and PMF with SAES data as input (PMF$_{SAES}$).

**Comments in detail:**

1. Lines 14-21: Abstract should be concise. These sentences can be improved to only briefly describe what necessary and importance of this study are, as well as to briefly address scientific question of this study.

**Response to Reviewer1 comment No. 1:** We have revised the abstract to make it more concise.

2. Line 19: Measurements at the two different sampling sites were conducted in this work, which should be addressed here. This is a very important information in PMF analysis. Aerosol chemical composition, especially organics, can be substantially different at different sites even in a same city region. Again, I don t believe this methodology is reasonable unless the authors can prove it.

**Response to Reviewer1 comment No. 2:** The rationality of the combination of data from two sites, as detailed above, is now added in the supporting information (Text S1). The following sentences are added into the main text.

**Lines 18-20:** "In this study, source apportionment of $PM_{2.5}$ using positive matrix factorization (PMF) model was conducted for urban Shanghai in the Yangtze River Delta region, China, utilizing a large suite of molecular and elemental tracers, together with water-soluble inorganic ions, OC and EC from measurements conducted at two sites from 9 November to 3 December, 2018."

3. Lines 21-22: This sentence can be improved. And what's the difference between molecular makers and organic makers here? Please rewrite this sentence.

4. Lines 23-24: Not clear. Please rewrite it.

**Responses to Reviewer1 comment No. 3 and No. 4:** The two terms are interchangeable. To avoid unnecessary confusion, we now adopt unified use of "molecular makers" in this article, and have replaced all "organic makers" with "molecular makers ". The sentence of concern is revised as below:

**Lines 21-24:** "The PMF analysis with inclusion of molecular makers (i.e. MM-PMF) identified 11 pollution sources, including three secondary source factors (i.e. secondary sulfate factor, secondary nitrate factor, and secondary organic aerosol (SOA) factor) and eight primary sources (i.e. vehicle exhaust, industrial emission/tire wear, industrial emission2, residual oil combustion, dust, coal combustion, biomass burning, and cooking)."

5. Line 24: The three -> These three. Another question, why did the authors select those sources for the quantification discussion here? Why not others included, since you have identified many source factors?

**Response to Reviewer1 comment No. 5: "**The three -> These three**"** revised as suggested. (Line28)

The three sources were extracted only by MM-PMF. Without organic markers, they are mixed with other sources and cannot be separated.

**Lines 26-29:** "Traditional PMF analysis relying on major components alone ($PMF_t$) was unable to resolve three organics-dominated sources (i.e. biomass burning, cooking, and SOA source factor). Utilizing organic tracers, the MM-PMF analysis determined that these three sources combined accounted for 24.4% of the total $PM_{2.5}$ mass."

6. Line 25: It has already been well known about the secondary source is an important source of atmospheric pollution in eastern China. Better to rewrite this sentence.

**Response to Reviewer1 comment No.6:** The sentence is revised as below.

**Lines 24-25**: "The secondary sources contributed 62.5% of the campaign-average $PM_{2.5}$ mass, with the secondary nitrate factor being the leading contributor."

7. Lines 26-27: It's a confusing sentence (Grouping different sources . . ., respectively). Please rewrite it. And what is the difference between SOA and SOC mentioned here together? Maybe only needing to show any individual of the two terms in the abstract section, if you don t need to explain the different things between SOA and SOC. In addition, SOC and POC should be defined before using them.

**Response to Reviewer1 comment No.7:** In order to make the summary more concise and highlight the important content, we choose to delete this part of the discussion. The abbreviations (SOA, SOC, and POC) are now defined in the body of the article. Specifically, SOA refers to secondary organic aerosols and SOC refers to secondary organic carbon.

8. Lines 33-35: Maybe a region scale should be given here, for instance, e.g., in eastern Asia or China?

**Response to Reviewer1 comment No.8**: revised as below.

**Line 39:** "…with the increasingly prominent problem of air quality in China, …"

9. Lines 35-36: This sentence should be revised to be better read. In addition, Zhang et al., 2009;). -> Zhang et al., 2009).

**Response to Reviewer1 comment No.9:** revised as below.

**Lines 40-41:** "Identifying the pollution sources and quantifying their contributions to ambient $PM_{2.5}$ pollution are of fundamental importance for PM emission reduction and air quality improvement (Chen et al., 2007; Zhang et al., 2009a)."

10. Lines 36-42: These sentences should be revised. As written here, aerosol climate and environmental impacts are not well described. They can be improved better for easy follow. For example, the sentence (lines 38-40) is discussing about climate impact of aerosols", while in the hereafter sentence, you conclude importance for air quality". I mean, in this paper, it may be not needed to highlight aerosol climate aspects in too much detail, but it can be discussed with a very brief version. In addition, in the first sentence (lines 33-35), you already have addressed the situation of PM2.5 pollution and it's important. But, in the sentences (lines 40-42), you highlighted the similar information again. Overall, I would suggest that more concise sentences can be made here to better describe importance and necessity that are associated with the work of the present study.

**Response to Reviewer1 comment No.10:** The whole paragraph has been revised for better clarity.

**Lines 37-39:** "Airborne $PM_{2.5}$ (i.e. particulate matter with aerodynamic diameter less than 2.5 μm) has attracted increased global attention due to its impact on climate, visibility, and human health (Chow et al., 2004; Liu et al., 2016; Foley et al., 2010)."

11. Lines 43-44: Again, this information has been declared in the above paragraph, no? Is the sentence of lines 44-45 expressing a similar knowledge as the lines 35-36?

**Response to Reviewer1 comment No.11:** We have deleted the repeated information in this paragraph.

12. Lines 46-47: What are ". . . other methods"? They could be briefly discussed here, since you would highlight some advantages of PMF model, especially you will need apply it with a unique PMF input

dataset.

**Response to Reviewer1 comment No.12:** revised as below.

**Lines 44-46:** "Compared with other methods, such as chemical mass balance (CMB) and multi-linear engine (ME-2), positive matrix factorization (PMF) (Paatero & Tapper, 1994) does not need to input the source profiles, and is able to provide as model outcome both the source profiles and contributions of various sources (Wang et al., 2018; Zhou et al., 2019)."

13. Lines 48-52: It may be not only due to this reason (as you discussed above) why PMF has been widely applying in aerosol source apportionment, as reported by many previous studies. Meanwhile, it would not be necessary to list too many references here. If I understood well, the authors would like to highlight the advantage of PMF compared to other receptor models, and why PMF was applied in the present study.

**Response to Reviewer1 comment No.13:** Revised as suggested. Since this paper does not focus on the advantages of different receptor models, we have removed unnecessary information and reduced reference numbers.

14. Line 55: Filter measurements can be also available for short-term source apportionment study. Not sure if this sentence is necessary/clear here with the current way.

**Response to Reviewer1 comment No.14:** Revised as below to improve clarity.

**Lines 51-53:** "Sample sizes of over a hundred could be acquired within a short time span in the order of a week, thus providing opportunities to study pollution source variations for short-term time windows."

15. Lines 58-62: These sentences are not clear enough. What are advantages of the observation methods in the present study, compared to AMS techniques? One could argue that the combination between AMS and organic tracers could be also a very good strategy for source apportionment of organic aerosols (e.g., Huang et al., 2014 Nature).

**Response to Reviewer1 comment No.15:** AMS/ACSM mass fragmentation ions are not unique source tracers, as the same fragment ions can come from different parent molecules (as stated in Lines 56-57). In comparison, the organic molecular markers are more source specific and allow us to more effectively separate organic sources.

**Lines 57-61:** "In comparison, molecular markers alleviate such ambiguity, therefore could significantly improve our ability in source identification and quantification. The recently commercialized Thermal desorption Aerosol Gas chromatography-mass spectrometry (TAG) system (Williams et al., 2014; Zhao et al., 2013b; Isaacman et al., 2014) has enabled acquiring hourly data of individual molecular markers, providing opportunities for more refined source apportionment."

Yes, we agree that the combination of AMS and organic markers will be a good strategy for source apportionment of organic aerosols. In the future, we plan to deploy TAG and AMS side-by-side to maximize our ability in resolving OA sources.

16. Lines 63-64. One transmission sentence between the two sentences is needed, otherwise one could understand that many previous studies performed in Shanghai could be only due to a large population. I guess you would express that Shanghai is a megacity, representing for a typical economic zone in China and with a large population there, while air pollution issues are complex due to unclear enough aerosol

sources that have to be better understood to further improve air quality.

17. Lines 64-71: Following the similar comments above, I would suggest the author to improve this paragraph.

**Response to Reviewer1 comment No.16 and No. 17:** We have rewritten the whole paragraph as below.

**Lines 62-69**: "Shanghai, a megacity with a population of 24.3 million and a total area of 6,340 km$^2$, represents a typical economic zone in China. Air pollution issues in Shanghai are complex and our knowledge of its aerosol sources still fall short of being sufficiently quantitative or comprehensive. Past source apportionment studies of PM$_{2.5}$ in Shanghai are either based on offline filter-based data that are inherently of low time-resolution (Du et al., 2017; Chang et al., 2018), or emissions-based numerical models (Li et al., 2015; Shu et al., 2019; Li et al., 2019; Feng et al., 2019). PM$_{2.5}$ source apportionment studies using online data as inputs, however, so far have been limited to the major aerosol species (i.e. inorganic ions, carbonaceous components and elements) (Wang et al., 2018), preventing proper separation of aerosol sources dominated by organic compounds."

18. Lines 67-68: It is not true that offline techniques are limited in air pollution event studies.

**Response to Reviewer1 comment No.18:** suggestion taken. We have deleted this loose description.

19. Line 77. As I mentioned above, the reasonable application to combine the two data sets from the two different sampling sites for PMF analysis needs to be further proved.

**Response to Reviewer1 comment No.19:** The comment has been addressed in our response to the general comments in the beginning of this response document.

20. Line 95. More detail in methods of organic compounds data analysis (including more descriptions in instrumental information) and data quality control should be described here, since such data set is a major part in the present study.

**Response to Reviewer1 comment No.20:** We've added one paragraph about TAG measurement in the manuscript.

**Lines 96-107:** "Quantification of hourly speciated organic markers were achieved using a Thermal desorption Aerosol Gas chromatography-mass spectrometry (TAG) (Aerodyne Research Inc., https://www.aerodyne.com/wp-content/themes/aerodyne/fs/TAG_0.pdf). The operation details and data quality are described in a separate paper (Wang et al., 2020), and only a brief description will be presented here. Briefly, ambient air was drawn through a PM$_{2.5}$ cyclone, then the sampled air was collected after passing through a carbon denuder to remove the gas phase and only particles were collected onto the collection matrix. The organics were then desorbed and transferred from the collection matrix to the GC column, with in-situ derivatization of the polar organics under a variable stream of saturated helium with derivatization agent (N-methyl-N-(trimethylsilyl) trifluoroacetamide). After GC column separation the target organics entered the MS chamber for analysis. It should be noted that with the current TAG instrument set-up, one hourly sample was collected at every odd hour, thus generating 12 hourly samples in a 24-h cycle. The post-sampling steps, including in-situ derivatization, thermal desorption, and GC/MS analysis, took ~1.5 h, and the next sampling started concurrently with the GC/MS analysis step, lasting for a full hour."

21. Line 131: 1/2 can be understood by "1 or 2" or "0.5". Same comment on line 132: "5/6".

**Response to Reviewer1 comment No.21:** revised as suggested. "5/6"-> " five-sixths ". (Line 150)

22. Line 134. A refence or acknowledgement when you use trajectory data and tool should be given.

**Response to Reviewer1 comment No.22:** References have been added as suggested.

**Lines 153-157:** "Backward trajectories of 36-h duration and arriving at an altitude of 100 m above ground level (AGL) over the PD site were calculated deploying the 0.5° Global Data Assimilation System (GDAS) meteorological data (https://www.ready.noaa.gov/archives.php). The trajectories were then classified into different clusters according to the geographical origins and movement process of the trajectories using the TrajStat model (Zhang et al., 2020)."

23. Lines 142-143. Some references should be given to support this statement. And what the link between this sentence and the ones hereafter? It seems that further discussion on potential impacts of the meteorological conditions on PM2.5 pollution observed in the present study is needed. Wind-dependent analysis of different aerosol sources can be applied here, I think.

**Response to Reviewer1 comment No. 23:** the sentence has been removed from the context.

The impact of meteorological conditions on PM$_{2.5}$ will be discussed in the episodic analysis part in section 3.3, including some wind analysis. This paragraph mainly gives a brief introduction of the PM pollution characteristics during the sampling period. As we have performed clustering analysis, we will not focus on additional wind-dependent analysis, which may distract the readers' attention.

24. Line 167: Define the terms BS, DISP and BS-DISP.

**Response to Reviewer1 comment No.24:** suggestion taken. The descriptions of the three error estimation methods have been added in Text S2.2 in the supporting information.

25. Line 169: Please show specific results and/or further prove the cases when the factors were mixed based on those base runs.

**Response to Reviewer1 comment No.25:** We have added the specific examples of the factor mixing in the base run results.

**Lines 195-198:** "The base run results show certain degrees of factor mixing, such as ~20% of biomass burning tracers-levoglucosan and mannosan- were mixed with secondary nitrate factor. Subsequently, a constrained run was performed to constrain levoglucosan and mannosan to be only present in the biomass burning factor (Wang et al., 2017)."

26. Lines 172-173: Figure 3 presents results resolved by the MM-PMF analysis. This information should be given in this sentence, since two PMF methods have been applied and hereafter you did a lot of discussion on Figure 3.

**Response to Reviewer1 comment No.26:** revised as suggested.

**Lines 200-204:** "The factor profiles of the 11-factor constrained run of MM-PMF are shown in Fig. 3, together with the time series of contributions from individual source factors. The diurnal variations of individual factor contributions are shown in Fig. 4. In summary, three secondary sources are resolved, namely, secondary sulfate factor, secondary nitrate factor, and SOA factor. Eight primary sources are resolved, and they are vehicle exhaust, industrial emission/tire wear, industrial emission2, residual oil combustion, dust, coal combustion, biomass burning, and cooking."

27. Line 173: Secondary nitrate factor (F1) was identified by high concentrations of nitrate and ammonium. . .". The PMF factor profile is described by contributions of each variable, but not their absolute concentrations. I think it might be enough when you directly discuss their contributions in factor profiles. Same comment on the similar statements.

**Response to Reviewer1 comment No.27:** revised as suggested. we have revised "concentrations" to "contributions". **(Line 207)**

28. Lines 177-179: what's difference between SOA and SOC? Please show results about . . . which can be seen from the correlation of the respective species with OC."

**Response to Reviewer1 comment No.28:** 'SOA' and 'SOC' have been defined in the article.

SOA (Secondary Organic Aerosols) is one of the sources of $PM_{2.5}$ in the air, as determined by the PMF model. When describing quantity of SOA, it has an air concentration unit of $\mu g/m^3$. And SOC is the secondary organic carbon, and has the concentration unit of $\mu gC/m^3$.

The sentence "…which can be seen from the correlation of the respective species with OC." is ambiguous and has been removed.

29. Lines 179-181: Those sentences could be read by: "Figure 4 presents the relative contributions of various source factors to PM2.5 and OC based on MM-PMF analysis, respectively. The F1 and F2 contributed 33.5% and 15.6% to the total PM2.5 mass concentration, respectively."

**Response to Reviewer1 comment No.29:** revised as suggested. **(see Lines 215-216)**

30. Lines 181-182: What does SOC contribute to, PM2.5 or OC?

**Response to Reviewer1 comment No.30:** The SOC contribute to total OC. We have revised the original statement.

**Lines 289-290:** "The SOC from the three secondary factors accounts for 48.6% (3.09 $\mu gC/m^3$) of the total OC mass on average across the whole study period."

31. Lines 182-183: Such sentences would be better discussed step-by-step for each factor, something like: "As shown in Fig. 5, the diurnal variation of the secondary nitrate factor shows high concentration during nighttime (e.g., 21:00 04:00), while its low concentration is observed during daytime (e.g., 11:00-14:00). etc. The F1, a secondary sulfate-rich factor, didn t present obvious diurnal pattern, which is however characterized by limited variations across the day. etc. . ." This is only an example. The authors don't need to exactly follow it, but clearer discussions are needed. In addition, influence of air temperature on the nitrate-rich factor should not be ignored, due to volatile properties of particulate ammonium nitrate.

**Response to Reviewer1 comment No.31:** Suggestion taken. The whole section has been revised to improve clarity. The impact of temperature on the diurnal variation of secondary nitrate has been added in the discussion.

**Lines 210-212:** "The higher contributions of secondary nitrate in the nighttime hours may be due to the lower nighttime temperature favoring the shifting of ammonium nitrate to the particle phase."

32. Line 186: please show evidence to further demonstrate influence of regional transportation on the F1, when you addressed it here.

**Response to Reviewer1 comment No.32:** the vague statement has been removed from the discussion.

33. Lines 187-189: the diurnal variations of RH and temperature can be plotted to support "high RH condition during the day". But it's not convinced to address that no obvious diurnal variation of sulfate-rich factor could be mainly due to the reason here. Other reasons should be further discussed. In addition, the conclusion of the sentence (lines 188-189) is weak, which is not sufficiently supported by the results.

**Response to Reviewer1 comment No 33.** Considering no convinced evidence can be found to support the statement, we now have removed the statement. Inferred from later backward trajectory analysis in section 3.2, the secondary sulfate showed obvious higher contributions under influence of northern continental and long-range transport air masses, indicating the regional transport feature of this factor.

**Lines 212-213:** "Contributions of F2 lack obvious diurnal patterns (Fig. 4), which may indicate the influence from regional transport, and this speculation is supported by the backward trajectory analysis in Sec. 3.2."

34. Lines 192-194: This sentence could be modified by "The factor 3 (F3) profile presents a high abundance of EC, OC, Ca, and Cu, as well as it contains some organic tracers (PAHs and organic acids, etc.), . . .". Lines 192-194: In addition, what does contribute to 11.3% of the total PM2.5 mass on average? An information to be needed to transfer this sentence to the next, to indicate this factor could be linked to vehicle exhaust. Otherwise it's hard to understand if this factor 3 is associated with vehicle exhaust source when you start to discuss it in the next sentence. What does this accounting for 16.2% to OC refer to, the total vehicle exhaust emissions or only POC?

**Response to Reviewer1 comment No.34:** revised as suggested. These sentences have been rewritten in the article and updated with the data of the model results.

**Lines 220-227**: "The third factor (F3) shows a high abundance of EC (38%), and is identified to be vehicle exhaust. It also contains high loadings of OC, Ca, and Cu, as well as some organic tracers (PAHs and organic acids) in the profile… F3 contributed to 12.6% of the total $PM_{2.5}$ mass and 19.4% of OC on average."

35. Lines 194-196: This sentence is not clear. Please revise it.

**Response to Reviewer1 comment No.35:** the whole paragraph has been re-written.

36. Line 197: what is daily variation?

**Response to Reviewer1 comment No.36**: Revised to be diurnal variation.

**Lines 224-225:** "The influence of vehicle exhaust on this factor is supported by the peak hours at 7:00-9:00 am and 5:00-7:00 pm in the diurnal variation (Fig. 4), coinciding with the morning and afternoon rush hours."

37. Lines 197-198: In addition, xxx has high correlation with NOx and CO, respectively, …? Mistakes in "R=67" and "R=58".

**Response to Reviewer1 comment No.37:** revised as suggested. This sentence has been rewritten and updated with the data of the model results.

**Lines 226-227:** "In addition, F3 has high correlations with $NO_x$ (R=0.68) and CO (R=0.48), further supporting the association of this factor with vehicle exhaust."

38. Lines 198-201: Needs to revise these sentences for better understanding.

**Response to Reviewer1 comment No.38:** The support for this sentence in the article was weak, and we have removed it from the discussion.

39. Line 204: loads -> loadings. Were you discussing on profile in this sentence? Please revise this sentence.

**Response to Reviewer1 comment No.39:** revised as suggested.

**Line 228:** "The profiles of the fourth factor (F4) contains high loadings of Fe and Mn."

40. Line 207: The F4 needs to be defined.

**Response to Reviewer1 comment No.40:** suggestion taken.

**Line 234:** "Therefore, F4 is considered as a mixed source of industrial emission and tire wear."

41. Lines 207-209: I think this sentence can be better improved.

**Response to Reviewer1 comment No.41:** The sentence is revised as below:

**Lines 229-231:** "These metals, together with Cu and Zn, are also reported by Pant & Harrison (2013) and Wang et al. (2018) to be associated with non-exhaust vehicle emissions such as tire wear."

42. Line 210: a mistake in R=49".

**Response to Reviewer1 comment No.42:** the typo mistake has been corrected.

**Lines 232-234**: "F4 shows a high correlation with $NO_x$ (R=0.49), and $NO_x$ in the Yangtze River Delta mainly originates from industrial and vehicular pollution sources (Fu et al.,2013)."

43. Lines 209-210: I don t understand why you discuss on F2 in the first phase of this sentence, then you get conclusion about F4?

**Response to Reviewer1 comment No.43**: This is a typo mistake and it should be "F4". (see updated revision in response to previous comment)

44. Line 210: source.The -> source. The". What does "the factor" refer to?

**Response to Reviewer1 comment No.44**: revised as suggested. "the factor" has been defined.

**Lines 234-235:** "Therefore, F4 is considered as a mixed source of industrial emission and tire wear. The contribution of this factor to total $PM_{2.5}$ mass and OC was minor, ..."

45. Lines 210-213: More discussions to be needed to further support "suggesting the benefit of the online high-resolution measurement.".

**Response to Reviewer1 comment No.45:** Below text is added.

**Lines 235-238:** "Industrial emission/tire wear could not be resolved as a separate source in the source apportionment analysis based on offline filter samples in this region (Du et al., 2017; Huang et al., 2014; Qiao et al., 2016). This inability is lifted with the hourly data, thus indicating the benefit of online high-time resolution measurements."

46. Line 213: "measurement.The" -> "measurement. The"

**Response to Reviewer1 comment No.46**: revised. **(Line 239)**

47. Lines 213-214: are often used in industrial processes -> are often used as industrial emission markers". But I am not sure if you did this meaning? In addition, the figure that you are discussing should be pointed out.

**Response to Reviewer1 comment No.47:** Cr and Ni have multiple emission sources and cannot be regarded as unique source tracers to industrial emissions. Chromium could come from solid waste incineration as well (Borai et al., 2002). Considering the urban locations of the sampling sites, waste incineration is unlikely an important source, thus we consider it as an industrial emission source. We revise as below to better explain the associated source for F5.

**Lines 239-241:** "The fifth factor (F5) is characterized by high loadings of Cr (74%), Ni (31%), and Zn (29%) (Fig. 3). Cr compounds are widely used in industrial activities such as plating, tanning, and metallurgy (Karar et al., 2006; Borai et al., 2002). In addition, this factor shows a strong correlation with CO (R= 0.68). Thus, it is regarded as industrial emission2."

48. Line 215: a mistake in "R=59".
**Response to Reviewer1 comment No.48:** see updated revision in response to previous comment.

49. Line 215: what did you discuss in this sentence "No diurnal variation was observed."? This is not clear at all.
**Response to Reviewer1 comment No.49:** It is revised as below to improve clarity.
**Line 242:** "... No diurnal variation is observed in this factor (Fig. 4)."

50. Line 215: "The factor" refer to the factor 5?
**Response to Reviewer1 comment No.50:** yes, it refers to the factor 5.
**Line 242:** "Factor contributions of F5 to total PM$_{2.5}$ and OC mass were minor, only 2.0% and 1.1%."

51. Lines 216-217: This sentence can be improved for better reading.
**Response to Reviewer1 comment No.51:** Revised as below to improve the clarity.
**Lines 244-245:** "The residual oil combustion factor (F6) is identified by high loadings of V (83%) and Ni (32%) (Fig.3). V is often used as a source tracer for residual oil combustion (Zhao et al., 2013c)."

52. Lines 217-218: how could you prove your factor 6 that mainly comes from ship transportation? Further discussion/evidence to be needed.
**Response to Reviewer1 comment No. 52:** The relevant supporting description has been added.
**Lines 245-247:** "The contributions of the residual oil combustion mainly come from ship transportation, due to the coastal geographical locations of Shanghai. The V/Ni ratio in the factor profile is 2.7, close to the ratio of fuel oil used in Shanghai Port (Zhao et al., 2013c)."

53. Line 218: again, this sentence is not clear: "The diurnal contribution at night was greater than that during daytime.". Please the authors carefully check such similar issues over the entire manuscript.
**Response to Reviewer1 comment No.53:** Revised as below. Also, the whole manuscript has been carefully examined and revised to improve clarity.
**Lines 247-248:** "The diurnal variation of this factor shows slightly higher concentrations during

nighttime (e.g., 21:00-23:00 and 03:00-07:00)."

54. Lines 219-220: to me "the contribution (10.7%)" is relatively high, but I don t think "is not negligible" is an appropriate use here.

**Response to Reviewer1 comment No.54:** Revised as suggested.

**Lines 248-250:** "F6 is a minor contributor to $PM_{2.5}$, accounting for 2.0%, while its contribution to OC is higher (7.1%). Therefore, residual oil combustion is also an important pollution source, especially to OM."

55. Lines 228-229: Mistakes in R=59 and R=56".

**Response to Reviewer1 comment No.55:** Revised.

**Lines 258-259:** "Good correlations with $SO_2$ (R= 0.68) and CO (R=0.68) further support the identification of this factor."

56. Line 230: "such as PAHs present in the source" -> "such as PAHs presented in the source profile"? or a figure you discussed should be linked here. "The results are different" -> "These results are different".

**Response to Reviewer1 comment No.56:** revised as below.

**Lines 259-261:** "No specific organic tracers such as PAHs are present in this source profile (Fig. 3). These results are different from those of Wang et al. (2017) and Yu et al. (2016), which may be attributed to regional differences in source profiles."

57. Lines 232-233: the two factors should be discussed separately.

**Response to Reviewer1 comment No.57:** suggestion taken. The two factors are discussed separately in Lines 262-274 (the text is copied below in the response to comment No. 58-60).

58. Lines 234-236: literatures should be cited here to support this statement. In addition, K needs to be defined.

**Response to Reviewer1 comment No.58:** We have added new references to support this statement.

**Lines 262-266:** "Levoglucosan and mannosan are uniquely emitted by biomass burning activities (Engling et al., 2006; Feng et al., 2013), thereby serving as reliable source tracers to indicate biomass burning in source analysis (Wang et al., 2019; Bond et al., 2007). In comparison, it is well documented that elemental potassium (K) suffers from potential interferential sources such as dust and fire work emissions (Yu et al., 2019)."

References:

Wang, Q. Q., Huang, X. H. H., Tam, F. C. V., Zhang, X. X., Liu, K. M., Yeung, C., Feng, Y. M., Cheng, Y. Y., Wong, Y. K., Ng, W. M., Wu, C., Zhang, Q. Y., Zhang, T., Lau, N. T., Yuan, Z. B., Lau, A. K. H., and Yu, J. Z.: Source apportionment of fine particulate matter in Macao, China with and without organic tracers: A comparative study using positive matrix factorization, Atmos. Environ., 198, 183-193, 2019.

Bond, T.C., Bhardwaj, E., Dong, R., Jogani, R., Jung, S., Roden, C., Streets, D.G., and Trautmann, N.M.: Historical emissions of black and organic carbon aerosols from energy-related combustion, 1850-2000, Global Biogeochem. Cy. 21, GB2018, 2007. https://doi.org/10.1029/2006GB002840.

Yu, S. Y., Liu, W. J., Xu, Y. S., Yi, K., Zhou, M., Tao, S., and Liu, W. X.: Characteristics and oxidative

potential of atmospheric PM2.5 in Beijing: Source apportionment and seasonal variation, Sci. Total. Environ., 650, 277-287, 2019.

59. Line 236: This factor, contributed 4.9% and 5.7% of total PM2.5 and OC mass on average, respectively. -> On average, this factor contributed 4.9% and 5.7% to the total PM2.5 and OC mass concentrations, respectively.
**Response to Reviewer1 comment No.59:** revised as suggested. **(Line 269)**

60. Line 237: biomass burning contains high-loading -> the source profile of biomass burning factor contains high loadings . . . (Fig.?).
**Response to Reviewer1 comment No.60:** revised as suggested.
**Lines 266-268:** "The source profile of biomass burning factor also contains high loadings of five-ring and six-ring PAHs that are considered to be derived from a mixed combustion sources (including coal combustion and biomass burning, etc.) (Fig. 3)."

61. Line 238: (including coal combustion, biomass burning, etc.) -> (including coal combustion and biomass burning, etc.)".
**Response to Reviewer1 comment No.61:** revised as suggested. (see response to previous comment)

62. Lines 239-241: To be suggested to discuss the two factors, biomass burning and cooking, separately.
**Response to Reviewer1 comment No.62:** revised as suggested. The two factors are now discussed separately in Lines 262-274.

63. Lines 242-243: which figure/table did you discuss in this sentence?
**Response to Reviewer1 comment No.63:** Here we discuss the content of figure 3.
**Lines 277-279:** "As shown in Fig.3, F11 is identied as a SOA factor on the basis of high loadings of toluene SOA tracer (2,3-dihydroxy-4-oxopentanoic acid), α-pinene SOA tracers (pinic acid and 3-methyl-1,2,3-butanetricarboxylic acid) and phthalic acid."

64. Line 243: This value 14.7% is not a major fraction, but it can be called as a relatively high fraction.
**Response to Reviewer1 comment No.64:** the statement has been revised.
**Line 284:** "The SOA factor accounts for 16.8% of PM$_{2.5}$, and 7.0% of OC on average (Fig. 5)."

65. Lines 244-245: Do the authors mean that this SOA factor could be associated with a mixture SOA factor from anthropogenic and biogenic sources? One could argue very small fraction of biogenic emission during wintertime in Shanghai region. In addition, it is not sufficient by the similar diurnal variations between nitrate and SOA to indicate that they have similar formation processes and interaction between anthropogenic-biogenic sources. Further discussion is needed to clearly explain such a factor and its potential formation processes.
**Response to Reviewer1 comment No. 65:** yes, based on the factor profiles, the SOA factor showed high loadings of both biogenic SOA tracers and anthropogenic SOA tracers. However, we can't simply rule out the biogenic SOA influence during wintertime. Several studies have reported the enhanced formation of biogenic SOA with the increased anthropogenic emissions (Ren et al., 2019). During the measurement period, we successfully measured two α-pinene SOA tracers (i.e. pinic acid, and 3-methyl-1,2,3-

butanetricarboxylic acid). The time series of the α-pinene SOA tracers showed very similar trend with toluene SOA tracer (i.e. DHOPA) and the two SOA tracers correlated well with total PM$_{2.5}$ (Figure R3).

[Figure]

**Figure R3.** Time series of PM$_{2.5}$ and measured α-pinene SOA tracers (sum of pinic acid, and 3-methyl-1,2,3-butanetricarboxylic acid) and toluene SOA tracer (i.e. DHOPA) during the sampling period.

The secondary nitrate factor and SOA showed similar temporal variations (R=0.63), especially under episodic hours (Figure R4). Thus, the statement was not only based on the similar diurnal variations. We agree that a stronger evidence is needed to affirm their potential formation processes, however, such exploration may be out of the scope of this manuscript, as here we mainly focus on the source apportionment of PM$_{2.5}$ using the molecular and elemental markers.

[Figure]

**Figure R4.** Time series of MM-PMF resolved source contributions to PM$_{2.5}$ from secondary nitrate and SOA factor during the sampling period.

References:
Ren, Y. Q., Wang, G. H., Tao, J., Zhang, Z. S., Wu, C., Wang, J. Y., Li, J. J., Wei, J., Li, H., and Meng, F.: Seasonal characteristics of biogenic secondary organic aerosols at Mt. Wuyi in Southeastern China: Influence of anthropogenic pollutants, Environ. Pollut., 252, 493-500, 2019.

66. Lines 258-264: those sentences should be discussed associated with your results when you need them to support something. Or if only for a summary purpose here, those contents could be moved into the introduction section.
**Response to Reviewer1 comment No.66:** we agreed with the reviewer and have removed this paragraph.

67. Line 270: It can be seen from Fig.7 that the composition varies with the source of the air mass. -> "As shown in Fig. 7, the PM2.5 sources vary evidently associated with different air mass origins."

**Response to Reviewer1 comment No.67:** revised as suggested (see updates in Lines 344-346).

**Lines 344-346:** "MM-PMF factor percentage contributions to $PM_{2.5}$ under each cluster during the sampling period are shown as pie charts in Fig. 7 and the mass contributions of individual factor under different clusters are shown in Fig. S10. The $PM_{2.5}$ sources vary evidently in their contributions under influences of air masses of different origins."

68. Lines 273-275: those sentences can be improved to be better read.
**Response to Reviewer1 comment No.68:** See the revised wording in Lines 335-338.
**Lines 335-338:** "Briefly, $PM_{2.5}$ concentration was the highest under influence of local air mass influence (i.e. cluster 2), with an average value of 67.7 $\mu g/m^3$, followed by the northeastern continental air mass (cluster 1, 59.1 $\mu g/m^3$). Lower PM concentrations were observed under influence of long-range transport air mass (cluster 4, 20.4 $\mu g/m^3$) and oceanic air mass (cluster 3, 30.0 $\mu g/m^3$)."

69. Line 276: cluster 1 is not associated with North China, but it could be more reasonable coming from eastern China.
**Response to Reviewer1 comment No.69:** suggestion taken. We have revised it to "northeastern" (line 329)

70. Lines 283 284: could be possible regional transport contributing to dust observed at this receptor site? Wind analysis together with dust source could help to further investigate this point.
**Response to Reviewer1 comment No.70:** we have removed this statement. From the cluster analysis, the dust factor showed slightly higher contributions under cluster 2, while lower and similar contributions under other clusters (Fig. S10), which may indicate influence by both local and regional sources. However, as dust is not an important PM source in the study period, we will not focus on this source. Additional wind analysis on this factor may distract the reader's focus.

R-19

[Figure]

**Figure S10.** Box plot of individual MM-PMF resolved source contributions for different clusters.

71. Lines 287-292: the clusters 3 and 4 could be associated with relatively clean air masses originated from ocean areas (see Fig. 6). The differences between clusters 3 and 4 are a long-range and a short-range transportation characterization, respectively. More locally-formed pollutants could be expected observed within cluster 3 air masses, while more regional pollutants could be linked to cluster 4.

**Response to Reviewer1 comment No.71:** We agree this analysis by the reviewer. Text is added in Lines 328-333.

**Lines 328-333:** "Four clusters are extracted based on the clustering analysis using the TrajStat model. Cluster 1 represents air mass originated from northeastern continental region, accounting for 17% of the toal trajectories. Cluster 2 is the local circulating air mass and account for 36% of the total trajectories. Cluster 3 (28% of total trajectories) and cluster 4 (20% of total trajectories) represent oceanic air mass and long-range transport air mass, respectively. Based on the mean trajectory length, more locally-formed pollutants were expected under clusters 2 and 3, while more regional transported pollutants could be linked to clusters 4 and 1."

72. Lines 297-299: Please rewrite this sentence.

**Response to Reviewer1 comment No.72:** The sentence is re-phrased as below.

**Lines 368-370:** "PM$_{2.5}$ concentrations higher than 75 μg/m$^3$ and lasting for more than 24 hours were determined as a PM episode in this study, and three episodes were extracted to examine the source compositions of PM$_{2.5}$ during different pollution periods (Fig. 2). "

73. Lines 299-300: Please revise this sentence for better reading.

**Response to Reviewer1 comment No.73:** The sentence is re-written.

74. Line 312: why the authors used the different HYSPLIT model versions in this paper? It should be declared in the main text as well.

**Response to Reviewer1 comment No.74:** in the revised manuscript, only one HYSPLIT model (i.e. TrajStat model) was used.

75. Lines 330-331: do the authors have any evidence to support this statement: "secondary sulfate factor may also be affected by primary emission of coal combustion."

**Response to Reviewer1 comment No.75:** considering no further evidence available for the statement, we have removed this sentence from the main text.

76. Lines 331-333: this sentence can be better improved.

**Response to Reviewer1 comment No.76:** The sentence has been re-written.

**Lines 391-392:** "EP2 and EP3, under the influence of local-circulating air mass, showed obvious higher contributions to $PM_{2.5}$ from secondary nitrate than EP1 (45-49% vs. 15%) (Fig. 8)."

77. Lines 333-335: give a link (e.g., figure or table) to your results what you are discussing.

**Response to Reviewer1 comment No.77:** suggestion taken, see updates in Lines 384-396.

78. Lines 335-336: a same comment as the above, give a link. In addition, which air mass trajectories are shorter, to support your statement the dust source is more affected by local and short-distance air mass transport."? It can be additionally/more directly helped to check wind dependence of your source factors (especially for local factors), in addition to air mass analysis.

**Response to Reviewer1 comment No.78:** see response to comment No.70, we have removed the statement in the main text that "dust source is more affected by local and short-distance air mass transport". The revised paragraph about the episodic analysis mainly focus on the major source contributors.

79. Lines 342-345: very weak evidence from the present study to support this sentence.

**Response to Reviewer1 comment No.79:** the ambiguous statement has been removed from the main text.

80. Lines 345-346: again, do you have evidence to support this? If yes, please show that.

**Response to Reviewer1 comment No.80:** the ambiguous statement has been removed from the main text.

81. Line 347: I guess this section might be better moved to methodology section or at beginning of section 3.1. Moreover, I guess it would be also worthful for doing correlations (with R and slope values) of individual component (e.g., nitrate, sulfate, ammonium, POC, SOC, EC, Ca2+, and K+) between the two different PMF analysis.

**Response to Reviewer1 comment No.81:** we have removed this section to Section 3.1.2 as "Impact of organic markers on source apportionment".

Actually, the two PMF runs share the same input data set except organic markers. Do you mean the correlations of the common PMF source factors from two PMF runs (i.e. Table 3)?

82. Lines 357-358: This sentence can be divided into two sentences to address the two different things. In addition, line 357 is shown -> are shown"; line 358 are show -> are shown.

**Response to Reviewer1 comment No.82:** revised as suggested.

**Line 299:** "The source profile and error estimation of the eight-factor solution of $PMF_t$ are shown in Fig. S7 and Text S2.

**Line 310:** "A comparison of individual factor contributions to $PM_{2.5}$ and OC between MM-PMF and $PMF_t$ is shown in Fig. 6."

83. Line 359: Table S7.Compare - >Table S7. Compare

**Response to Reviewer1 comment No.83:** revised.

84. Lines 384-402: I think this conclusion section needs a major revision to highlight the key findings and further atmospheric implications from the present study, in addition to some technical corrections what I have listed below.

**Response to Reviewer1 comment No.84:** the whole conclusion section has been revised. The atmospheric implication of this study has been added.

**Lines 418-424:** "This study has demonstrated with field observation data that the combination of online organic molecular markers and online elemental tracers and other PM major components provided more comprehensive characterization of the PM pollution sources, in particular those dominated by organics which would be otherwise mixed into other sources and bias source apportioned to these "other sources". The hourly resolution in source factor contributions allows convenient utilization of those hourly data that have been routinely measured or obtained (e.g., meteorological conditions, gas pollutants, and backward trajectories analysis) to achieve an in-depth understanding of the source origins. The high time resolution data also has enabled the examination of pollution characteristics of different short-term PM pollution episodes…"

85. Line 385: a typical city -> a megacity

**Response to Reviewer1 comment No.85:** revised **(Line 400).**

86. Line 388: what does contribution of secondary pollution sources contribute to?

**Response to Reviewer1 comment No.86:** This sentence is re-written to improve clarity.

**Lines 404-406**: "The three secondary sources combined (i.e. sum of the secondary nitrate, secondary sulfate and SOA factor) contributed to more than 60% of $PM_{2.5}$ mass and 48.6% of the total OC."

87. Line 392: Please rewrite this sentence.

**Response to Reviewer1 comment No.87:** revised as suggested. See response to previous comment.

88. Lines 400-402: Very common statements, since they are well-known knowledges and have been reported by many previous studies. As I commented above, the authors would be able to better introduce

what the key findings/implications supported by the present study.

**Response to Reviewer1 comment No.88:** revised as suggested.

**Lines 410-417:** "The backward trajectory clustering analysis on the MM-PMF resolved source contributions revealed the impact of the air mass origins on different source factors. Secondary nitrate showed much higher contributions under local air mass influence, while secondary sulfate showed higher contributions under the influence of northeastern continental and long-range transport air mass. Three episodic events occurred during the measurement period and our analysis showed enhanced contributions from secondary nitrate and SOA factors in episodic hours. Increased contribution from secondary sulfate was observed in the episode influenced by northeastern continental air masses. The results indicated that PM pollution in winter in Shanghai area is greatly affected by both local pollutant emissions and the regional transport from the northeastern continental regions."

Comments on figures:

89. Figure 1 should be improved by adding a larger scale map to indicate where Shanghai is located in China or eastern Asia.

**Response to Reviewer1 comment No.89:** revised as suggested. **(Figure 1)**

90. Figure 2 and Figure 8 should be merged together by a single figure. There is no need to repeat PM2.5 temporal variations again in the main text. If you wanted to highlight its relation with wind direction, you can add wind direction in Figure 2.

**Response to Reviewer1 comment No.90:** suggestion taken. **(Figure 2)**

91. In figure 3, I think it's better to draw the time series of those sources with lines than using solid-filled areas, since we cannot see clearly if there is a missing gap for the sampling.

**Response to Reviewer1 comment No.91:** revised as suggested. **(Figure 3)**

92. Figure 4: Again, without proving your reasonable PMF analysis using the data from different sites, it isn't convinced to accept these quantitative results.

**Response to Reviewer1 comment No.92:** This comment has been addressed in the **"Response: General comments"**

93. Figure 5: why did the authors show two-hour resolution diurnal variations instead of using hourly, since you have hourly data as mentioned in previous section? I suggest to use hourly.

**Response to Reviewer1 comment No.93:** The time resolution for major PM components is one hour, however, the measurement data of organic markers by TAG are only available for every odd hour. we have updated the statement in the main text.

**Lines 98-107:** "The operation details and data quality are described in a separate paper (Wang et al., 2019), and only a brief description will be presented here…It should be noted that with the current TAG instrument set-up, one hourly sample was collected at every odd hour, thus generating 12 hourly samples in a 24-h cycle. The post-sampling steps, including in-situ derivatization, thermal desorption, and GC/MS analysis, took 1.5 h, and the next sampling started concurrently with the GC/MS analysis step, lasting for a full hour."

94. Figure 6 and Figure 7: I believe that it can be more visualized to merge the two figures within a single

one, e.g., different clusters associated with different pie charts.

**Response to Reviewer1 comment No.94:** revised as suggested. The original figures 6 and 7 have been combined to form the new **Figure 7.**

95. Figure 9: a same as Figure 4.

**Response to Reviewer1 comment No.95:** For corresponding comments, please see **"Response: General comments".** **(Figure 8).**

96. Figure 11: The mass concentrations of those source factors can be also presented here to better understand the differences between the different PMF analysis methodologies.

**Response to Reviewer1 comment No.96:** suggestion taken. The mass contributions of the source factors have been added in the figure, besides the percentage contributions. **(Figure 6)**

---

## Author Response (AR2)

**Point-by-point response to comments by Editor**

We thank the editor for the detailed comments. We have revised the manuscript accordingly. Below is our point-by-point response to each comment, marked in blue. Changes made to the main text are also marked in blue in the revised manuscript file. By this chance, we would like to show our enormous respect to Dr. Willy for doing such a thorough job to improve our paper.

**For the Main text:**

1. Lines 21, 22, 23, 27, 37, 67, 80, 172, 306, 336, 375 (twice), 378, 401, 405, 406, and 407: Replace "i.e." by "i.e.,".

**Response:** revised as suggested. **(Line 21, 21, 22, 23, 27, 37, 67, 80, 172, 308, 337, 377 (twice), 380, 404, 408, 409, and 410)**

2. Line 30: Replace "biasing source" by "biasing the source" and replace "on MM-PMF" by "on the MM-PMF".

**Response:** revised as suggested. **(Line 30)**

3. Line 32: Replace "and SOA" by "and the SOA" and replace "to PM2.5" by "to the PM2.5".

**Response:** revised as suggested. **(Line 32)**

4. Line 37: Replace "particulate matters" by "particulate matter".

**Response:** revised as suggested. **(Line 37)**

5. Line 70: Replace "compositions," by "composition,".

**Response:** revised as suggested. **(Line 70)**

6. Line 78: Replace "Method" by "Methods".

**Response:** revised as suggested. **(Line 78)**

7. Line 80: Replace "compositions" by "composition".

**Response:** revised as suggested. **(Line 80)**

8. Line 86: Replace "Locations" by "Location".

**Response:** revised as suggested. **(Line 86)**

9. Line 90: Replace "by Monitor" by "by a Monitor".

**Response:** revised as suggested. **(Line 90)**

10. Line 93: Replace "nondispersive" by "energy-dispersive".

**Response:** revised as suggested. **(Line 93)**

11. Line 96: Replace "were achieved using a Thermal" by "was achieved using Thermal".

**Response:** revised as suggested. **(Line 96)**

12. Line 101: Abbreviations and acronyms, here "GC", should be defined (written full-out) when first used.

**Response:** revised as suggested. **(Line 101-102)**

13. Line 102: Replace "with derivatization" by "with a derivatization".

**Response:** revised as suggested. **(Line 103)**

14. Line 111: Replace "agreements in" by "agreement in".

**Response:** revised as suggested. **(Line 111)**

15. Line 128: Replace "specie in" by "species in".

**Response:** revised as suggested. **(Line 128)**

Line 132: Replace "of mass" by "of the mass".

**Response:** revised as suggested. **(Line 132)**

16. Line 137: Replace "for PMF" by "for the PMF".

**Response:** revised as suggested. **(Line 137)**

17. Line 141: Replace "to PMF" by "to the PMF".

**Response:** revised as suggested. **(Line 141)**

18. Line 142: Replace "anhydro sugars" by "anhydrosugars".

**Response:** revised as suggested. **(Line 142)**

19. Line 143: Replace "and etc." by "etc.".

**Response:** revised as suggested. **(Line 143)**

20. Line 154: Replace "path on" by "paths on".

**Response:** revised as suggested. **(Line 154)**

21. Line 164: Replace "contributing to" by "contributing with".

**Response:** revised as suggested. **(Line 164)**

22. Line 165: Replace "contributed to" by "contributed with" and replace "of PM2.5" by "of the PM2.5".

**Response:** revised as suggested. **(Line 165)**

23. Line 166: Replace "component and" by "components and".

**Response:** revised as suggested. **(Line 166)**

24. Page 8, Tables 1 and 2: Several numeric data are given with too many significant figures; two significant figures suffice and three in case the first one is "1"; thus, for example, replace "46.3" and "33.8" by "46" and "34", respectively, and replace "45.91" and "39.17" by "46" and "39", respectively; also within the text several numeric data are given with too many significant figures; for example, in line 164, replace "46.3±33.8" by "46±34" and replace "32.0% and 25.2%" by "32% and 25%", and in line 303, replace "0.843-0.993" by "0.84-0.99".

**Response:** revised as suggested. **(Page 8)**

25. Line 192: Replace "detail descriptions" by "detailed description".

**Response:** revised as suggested. **(Line 192)**

26. Line 194: Replace "for PMF" by "for the PMF".

**Response:** revised as suggested. **(Line 194)**

27. Lines 196 and 198: Replace "with secondary" by "with the secondary".

**Response:** revised as suggested. **(Line 196)**

28. Line 200: Replace "are given" by "is given".

**Response:** revised as suggested. **(Line 200)**

29. Line 208: Replace "Secondary nitrate" by "The secondary nitrate".

**Response:** revised as suggested. **(Line 208)**

30. Line 219: Replace "The individual" by "Individual".

**Response:** revised as suggested. **(Line 219)**

31. Line 228: Replace "contributed to" by "contributed with".

**Response:** revised as suggested. **(Line 228)**

32. Line 229: Replace "The profiles" by "The profile".

**Response:** revised as suggested. **(Line 229)**

33. Line 233: Replace "with the tire" by "with tire".

**Response:** revised as suggested. **(Line 233)**

34. Line 235: Replace "al.,2013" by "al., 2013".

**Response:** revised as suggested. **(Line 234-235)**

35. Line 236: Replace "to total" by "to the total".

**Response:** revised as suggested. **(Line 236)**

36. Line 248: Replace "in Shanghai" by "in the Shanghai" and replace "diurnal variations" by "diurnal variation".

**Response:** revised as suggested. **(Line 248)**

37. Line 254: Replace "to total" by "to the total".

**Response:** revised as suggested. **(Line 254)**

38. Line 258: Replace "factor ro" by "factor to".

**Response:** revised as suggested. **(Line 258)**

39. Line 259: Replace "Diurnal variations" by "The diurnal variation".

**Response:** revised as suggested. **(Line 259)**

40. Line 262: Replace "contributes to" by "contributes with".

**Response:** revised as suggested. **(Line 262)**

41. Line 267: Replace "of biomass burning factor also ontains" by "of the biomass burning factor also contains".

**Response:** revised as suggested. **(Line 267)**

42. Line 268: Replace "from a mixed" by "from mixed".

**Response:** revised as suggested. **(Line 268)**

43. Line 269: Replace "diurnal variations" by "diurnal variation".

**Response:** revised as suggested. **(Line 269)**

44. Line 272: Replace "diurnal variations of the cooking factor show" by "diurnal variation of the cooking factor shows".

**Response:** revised as suggested. **(Line 272)**

45. Line 274: Replace "contributes to only" by "contributes only".

**Response:** revised as suggested. **(Line 274)**

46. Line 278: Replace "Fig.3" by "Fig. 3" and replace "of toluene" by "of a toluene".

**Response:** revised as suggested. **(Line 278)**

47. Line 280: Replace "represents a mixed" by "represents mixed".

**Response:** revised as suggested. **(Line 280)**

48. Line 282: Replace "and secondary" by "and the secondary".

**Response:** revised as suggested. **(Line 282)**

49. Line 287: Replace "Overall, MM-PMF" by "Overall, the MM-PMF".

**Response:** revised as suggested. **(Line 287)**

50. Line 292: Replace "comonality" by "commonality".

**Response:** revised as suggested. **(Line 293)**

51. Line 294: Replace "accout for" by "account for".

**Response:** revised as suggested. **(Line 294)**

52. Line 295: Replace "contributed the" by "contributing the".

**Response:** revised as suggested. **(Line 295)**

53. Line 297: Replace "PMF model" by "The PMF model".

**Response:** revised as suggested. **(Line 297)**

54. Line 305: Replace "This diffince" by "This difference".

**Response:** revised as suggested. **(Line 305)**

55. Line 307: Replace "for secondary sulfate factor and EC for vehicle" by "for the secondary sulfate factor and EC for the vehicle".

**Response:** revised as suggested. **(Line 307)**

56. Line 311: Replace "than those for" by "than for".

**Response:** revised as suggested. **(Line 311)**

57. Line 313: Replace "contributions from" by "the contribution from".

**Response:** revised as suggested. **(Line 313)**

58. Line 315: Replace "MM-MMPF" by "MM-PMF".

**Response:** revised as suggested. **(Line 315)**

59. Line 317: Replace "distribute contributions" by "distribute the contributions".

**Response:** revised as suggested. **(Line 317)**

60. Line 318: Replace "between two" by "between the two".

**Response:** revised as suggested. **(Line 318)**

61. Line 319: Replace "Coal combution contribution" by "The coal combustion contribution" and replace "and residual" by "and the residual".

**Response:** revised as suggested. **(Line 319)**

62. Line 320: Replace "generates more" by "generates a more".

**Response:** revised as suggested. **(Line 320)**

63. Line 321: Replace "In another" by "In other".

**Response:** revised as suggested. **(Line 321)**

64. Line 325: Replace "contribution to" by "contributions to".

**Response:** revised as suggested. **(Line 325)**

65. Line 328: Replace "Fig.7" by "Fig. 7".

**Response:** revised as suggested. **(Line 328)**

66. Line 329: Replace "mass originated from northeastern" by "masses originating from the northeastern".

**Response:** revised as suggested. **(Line 329)**

67. Line 330: Replace "of the toal trajectories" by "of all trajectories" and replace "account for 36% of the total" by "accounts for 36% of all".

**Response:** revised as suggested. **(Line 330)**

68. Line 331: Replace "of total trajectories" by "of all trajectories" (twice) and replace "air mass and" by "air masses and".

**Response:** revised as suggested. **(Line 331)**

69. Line 332: Replace "air mass," by "air masses,".

**Response:** revised as suggested. **(Line 332)**

70. Line 355: Replace "Briefly, PM2.5" by "Briefly, the PM2.5".

**Response:** revised as suggested. **(Line 335)**

71. Line 336: Replace "air mass influence" by "air masses".

**Response:** revised as suggested. **(Line 336)**

72. Line 337: Replace "air mass" by "air masses".

**Response:** revised as suggested. **(Line 337)**

73. Line 338: Replace "air mass" by "air masses" (twice).

**Response:** revised as suggested. **(Line 338)**

74. Line 341: Replace "by TrajStat model. The percentage in the brackets" by "by the TrajStat model. The percentage in parentheses".

**Response:** revised as suggested. **(Line 341-342)**

75. Line 344: Replace "MM-PMF factor" by "The MM-PMF factor".

**Response:** revised as suggested. **(Line 344)**

76. Line 345: Replace "of individual factor" by "of the individual factors".

**Response:** revised as suggested. **(Line 345)**

77. Line 346: Replace "influences of air" by "influence of air".

**Response:** revised as suggested. **(Line 346)**

78. Line 347: Replace "and vehicle" by "and the vehicle".

**Response:** revised as suggested. **(Line 347)**

79. Lines 347-348: Replace "regardless air mass cluster types. Secondary nitrate" by "regardless of air mass cluster type. The secondary nitrate".

**Response:** revised as suggested. **(Line 347-348)**

80. Line 349: Replace "ofNOx" by "of NOx".

**Response:** revised as suggested. **(Line 349)**

81. Line 353: Replace "need from heating" by "need of heating".

**Response:** revised as suggested. **(Line 353)**

82. Line 356: Replace "from northern continental area. Vehicle" by "from the northern continental area. The vehicle".

**Response:** revised as suggested. **(Line 356)**

83. Line 362: Replace "Dust show similar" by "Dust shows similar".

**Response:** revised as suggested. **(Line 362)**

84. Line 374: Replace "mainly fall under" by "mainly falls under".

**Response:** revised as suggested. **(Line 374)**

85. Line 377: Replace "than the" by "than during the".

**Response:** revised as suggested. **(Line 377)**

86. Line 388: Replace "among sources while source" by "among the sources while the source".

**Response:** revised as suggested. **(Line 389)**

87. Line 391: Replace "in Beijing" by "in the Beijing".

**Response:** revised as suggested. **(Line 392)**

88. Line 393: Replace "high contributions of" by "high contribution of".

**Response:** revised as suggested. **(Line 394)**

89. Line 394: Replace "among three" by "among the three".

**Response:** revised as suggested. **(Line 395)**

90. Line 413: Replace "air mass" by "air masses".

**Response:** revised as suggested. **(Line 414)**

Line 416: Replace "in Shanghai" by "in the Shanghai".

**Response:** revised as suggested. **(Line 417)**

91. Line 426: Replace "are suggested" by "is suggested".

**Response:** revised as suggested. **(Line 427)**

92. Pages 21-25, References:

- for authors with more than one initial there should be a space between the initials;

- titles of journal articles should be in lower case and not in Title Case.

Line 475: Replace "205 226" by "205-226".

Line 550: Replace "Acs. Earth. Space. Chem." by "ACS Earth Space Chem.".

Lines 556-557: Replace "ACS Earth and Space Chemistry" by "ACS Earth Space Chem.".

Line 573: Replace "Kong, china" by "Kong, China".

**Response:** References have been carefully checked and revised as suggested. **(Line 437-588)**

**For the Supplement:**

S1. Line 8: Replace "from PD site" by "from the PD site".

**Response:** revised as suggested. **(Line 8)**

S2. Line 9: Replace "from SAES site" by "from the SAES site".

**Response:** revised as suggested. **(Line 9)**

S3. Line 17: Replace "for general" by "for the general".

**Response:** revised as suggested. **(Line 17)**

S4. Line 20: Replace "are similar." by "are similar".

**Response:** revised as suggested. **(Line 21)**

S5. Line 27: Replace "in Pudong district" by "in the Pudong district".

**Response:** revised as suggested. **(Line 28)**

S6. Line 28: Replace "in Xuhui district" by "in the Xuhui district".

**Response:** revised as suggested. **(Line 29)**

S7. Line 30: Replace "are similar." by "are similar".

**Response:** revised as suggested. **(Line 31)**

S8. Lines 32, 63, and 71: Replace "i.e." by "i.e.,".

**Response:** revised as suggested.

S9. Line 35: Replace "of PM2.5" by "of the PM2.5".

**Response:** revised as suggested. **(Line 36)**

S10. Line 37: Replace "at two" by "at the two".

**Response:** revised as suggested. **(Line 38)**

S11. Line 42: Replace "in sync" by "in synchrony".

**Response:** revised as suggested. **(Line 43)**

S12. Line 43: Replace "at SAES site" by "for the SAES site".

**Response:** revised as suggested. **(Line 44)**

S13. Line 49: Replace "well between them throughout" by "well throughout".

**Response:** revised as suggested. **(Line 50)**

S14. Line 51: Replace "between two" by "between the two".

**Response:** revised as suggested. **(Line 51)**

S15. Line 55: Replace "combined data set to achieve comprehensive" by "a combined data set to achieve a comprehensive".

**Response:** revised as suggested. **(Line 55-56)**

S16. Line 57: Replace "at PD" by "at the PD".

**Response:** revised as suggested. **(Line 58)**

S17. Lines 64 and 68: Replace "at two" by "at the two".

**Response:** revised as suggested. **(Line 64, 67, 68)**

S18. Line 74: Replace "between PD" by "between the PD".

**Response:** revised as suggested. **(Line 75)**

S19. Line 75: Replace "during measurement" by "during the measurement".

**Response:** revised as suggested. **(Line 76)**

S20. Line 78: Replace "between PD" by "between the PD".

**Response:** revised as suggested. **(Line 79)**

S21. Line 79: Replace "during measurement" by "during the measurement".

**Response:** revised as suggested. **(Line 80)**

S22. Line 84: Replace "at PD and SAES sites during" by "at the PD and SAES sites during the".

**Response:** revised as suggested. **(Line 85-86)**

S23. Page S-5, Table S1: Several numeric data are given with too many significant figures; two significant figures suffice and three in case the first one is "1".

**Response:** revised as suggested. **(Table S1)**

S24. Line 91: Replace "during measurement" by "during the measurement".

**Response:** revised as suggested. **(Line 92)**

S25. Line 97: Replace "Change of" by "The change of", replace "are often" by "is often", and replace "help factor" by "help in the factor".

**Response:** revised as suggested. **(Line 98)**

S26. Line 103: Replace "examining factor" by "examining the factor".

**Response:** revised as suggested. **(Line 104)**

S27. Lines 104-105: Replace "from secondary nitrate factors" by "and the secondary nitrate factor".

**Response:** revised as suggested. **(Line 105-106)**

S28. Line 109: Replace "Finally," by "Finally, the".

**Response:** revised as suggested. **(Line 111)**

S29. Line 119: Replace "in EPA" by "in the EPA".

**Response:** revised as suggested. **(Line 120)**

S30. Line 124: Replace "there are a" by "there is a".

**Response:** revised as suggested. **(Line 125)**

S31. Line 129: Replace "indicate robust" by "indicates robust".

**Response:** revised as suggested. **(Line 130)**

S32. Line 136: Replace "of BS" by "of the BS".

**Response:** revised as suggested. **(Line 137)**

S33. Lines 141 and 146: Replace "The model" by "Model".

**Response:** revised as suggested. **(Line 142, 147)**

S34. Line 156: Replace "in eight-factor" by "in the eight-factor".

**Response:** revised as suggested. **(Line 157)**

S35. Lines 164 and 165: Replace "The average concentration of PM2.5 and its major components under different clusters" by "Average concentration of PM2.5 and its major components for different air mass clusters".

**Response:** revised as suggested. **(Line 165-166)**

S36. Line 168: Replace "under different clusters" by "for different air mass clusters".

**Response:** revised as suggested. **(Line 169)**

S37. Line 177: Replace "The average concentrations" by "Average concentrations".

**Response:** revised as suggested. **(Line 178)**

---

## Author Response (AR3)

**Point-by-point response to comments by Editor**

We thank the editor for the detailed comments. We have revised the manuscript accordingly. Below is our point-by-point response to each comment, marked in blue. Changes made to the main text are also marked in blue in the revised manuscript file.

**For the Main text:**

1. Lines 101-102. Replace "gas chromatography spectrometer (GC)" by "gas chromatography (GC) spectrometer".

**Response:** revised as suggested. **(Line 101-102)**

2. Page 8, Table 1, for Cl-: Replace "0.8" by "0.78" and replace "0.5" by "0.52".

**Response:** revised as suggested. **(Page 8, Table 1)**

Besides adjustments requested by the Editor, we also checked the manuscript carefully for typos, etc. Corrections are made as follows:

1. Line 103, inserted "," after "separation".
2. Line 300, replaced "PMFt" by "$PMF_t$".
3. Line 401, replaced "megacity city" by "megacity".